

# REMA reveals spatial variability within the Dotson Melt Channel

Ann-Sofie Priergaard Zinck[1,2], Bert Wouters[1,2], Erwin Lambert[3], and Stef Lhermitte[4,1]

[1]Delft University of Technology, Delft, The Netherlands
[2]Utrecht University, Utrecht, The Netherlands
[3]Royal Netherlands Meteorological Institute (KNMI), De Bilt, The Netherlands
[4]Katholieke Universiteit Leuven, Leuven, Belgium

**Correspondence:** Ann-Sofie P. Zinck (a.p.zinck@tudelft.nl)

**Abstract.** The intrusion of warm circumpolar deep water causes ice shelves in the Amundsen and Bellinghausen Sea Embayments of Antarctica to melt from below, thereby potentially putting their stability at risk. Earlier studies have shown how digital elevation models can be used to obtain high-resolution ice shelf basal melt rates. However, there has been limited availability of high-resolution elevation data, a gap the Reference Elevation Model of Antarctica (REMA) has filled. In this study we use a novel combination of REMA and CryoSat-2 elevation data to obtain high-resolution basal melt rates of the Dotson Ice Shelf in a Lagrangian framework, at a 50m spatial resolution on a 3-yearly temporal resolution. We present a novel Basal melt rates Using REMA and Google Earth Engine (BURGEE) method, which high resolution is supported through a sensitivity study assuming that the quality of the Lagrangian displacement is the main error source. The high-resolution basal melt rates show a good agreement with an earlier 500-m resolution basal melt product based on CryoSat-2, with a wide melt channel extending from the grounding line to the ice shelf front. Our high-resolution product indicates that the pathway and spatial variability of the main channel is influenced by a pinning point on the ice shelf. Additionally, it reveals a narrower coastal channel running along the western margin, which was not identified in the lower-resolution melt product, but does show up in a recent ocean modeling study. This emphasizes the importance of high-resolution basal melt rates to expand our understanding of channel formation and melt patterns. BURGEE can be expanded to a Pan-Antarctic study of high-resolution basal melt rates. This will provide a better picture of the (in)stability of Antarctic ice shelves.

## 1 Introduction

Ice shelves in the Amundsen Sea Embayment of Antarctica are subject to intrusion of warm circumpolar deep water which is one of the processes that can cause basal melting (Noble et al., 2020). This can lead to ice shelf thinning, grounding line retreat and a reduction in the ice shelf resistive forces on the tributary glaciers (Schoof, 2007). The thinning and force reduction put the tributary glaciers at risk, particularly in regions with a retrograde bed slope where marine ice sheet instability processes might be initiated (Schoof, 2007; Ritz et al., 2015). Therefore, it is important to monitor basal melting and ice shelf thinning to gain knowledge about the potential consequences. Monitoring can be done in-situ from ice penetrating radar (Berger et al., 2017; Lindbäck et al., 2019), phase sensitive radars (Lindbäck et al., 2019; Vaňková and Nicholls, 2022) or direct ocean measurements of conductivity and temperature at depth (Vaňková and Nicholls, 2022) or remotely through satellite observations of changes



in ice shelf surface elevation (Berger et al., 2017; Adusumilli et al., 2020). The in-situ measurements can provide melt and thinning rates at a high accuracy, but they are usually restricted to a few point measurements and with a temporal resolution defined by field work constrains. Remote sensing observations, on the other hand, can provide high spatial and temporal resolution but come with a series of assumptions needed to turn surface elevation measurements into thinning and melt rates.

Previous studies have shown how various satellite techniques can be used to obtain ice shelf thinning and basal melt rates
(Rignot et al., 2013; Adusumilli et al., 2020; Shean et al., 2019; Berger et al., 2017; Gourmelen et al., 2017). This can be done by using, e.g., stereo imagery (Shean et al., 2019), synthetic aperture radar (Berger et al., 2017), altimetry (Rignot et al., 2013; Moholdt et al., 2014; Gourmelen et al., 2017), or by a combination of the different techniques (Shean et al., 2019; Adusumilli et al., 2020). Common to all remote-sensing-based basal melt rate products is that they assume hydrostatic equilibrium to translate remotely sensed surface elevations into ice thickness, from which basal melt rate estimates can be obtained through
a mass conservation approach. This is often done in a Lagrangian framework where the basal mass balance of an ice parcel is assessed in contrast to the Eulerian framework where the basal mass balance of a given point in space is assessed. Applying a Lagrangian framework thus allows one to assess the thinning and basal melt rate of a given ice parcel over time and takes the ice flow into consideration. Berger et al. (2017) was one of the first studies providing high resolution Lagrangian basal melt rates of an Antarctic ice shelf. They used surface elevations based on satellite imagery from the twin synthetic aperture radar
satellite mission TanDEM-X, from which digital elevation models (DEMs) were generated by co-registering the TanDEM-X elevations to a CryoSat-2 DEM (Helm et al., 2014). This approach allowed to assess basal melt rates of the Roi Baudoin Ice Shelf at a 10 m spatial resolution, and thereby also revealing several small-scale melt channels. Shean et al. (2019) used stereo imagery from the WorldView and GeoEye satellites to create digital surface models of the Pine Island Glacier Ice Shelf, from which high-resolution DEMs were generated by co-registering to laser altimetry measurements from ICESat and NASA
Operation IceBridge. The resulting DEMs from 2008 to 2015 were used to obtain 32-256 m multi-scale resolution basal melt rates. Gourmelen et al. (2017) took on a different approach by only using altimetry measurements. They used CryoSat-2 swath measurements to obtain 500 m resolution melt rates of the Dotson Ice Shelf over the period from 2010-2016. They revealed a ∼5 km wide channel extending from the area around the grounding zone all the way to the ice shelf front. There is, however, one clear limitation to relying fully on satellite altimetry measurements, which is the fact that mountainous terrain near the ice
shelf margins prevents the satellite radar signal from reaching all parts of the ice shelf. From Fig. 1 it is clear that this is an issue on Dotson, which also becomes evident in the spatial coverage of the basal melt rates obtained in Gourmelen et al. (2017). For example the Kohler grounding zone (see Fig. 1) is poorly constrained, due to the surrounding topography limiting the CryoSat-2 coverage on the ice shelf, although here high melt rates are to be expected due to the intrusion of warm circumpolar deep water into the ice shelf cavity (Jacobs et al., 1992).

One drawback of the existing methods is the varying spatial/temporal resolution. Furthermore, the duration of the study periods alongside the temporal resolution of the results also vary. Both the chosen temporal and spatial resolution will put a constraint on the information level of the resulting basal melt rates since the basal melt pattern may vary on seasonal to inter-annual time scales (Watkins et al., 2021; Wearing et al., 2021; Dutrieux et al., 2013; Stanton et al., 2013). It is therefore



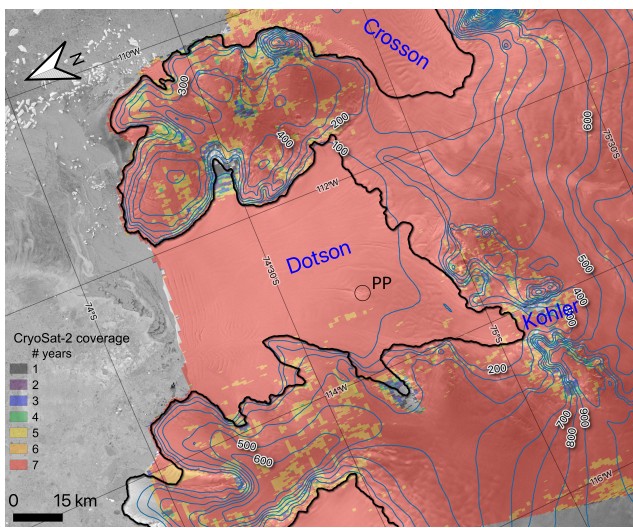

**Figure 1.** Overview of the test site with the Dotson and Crosson ice shelves marked alongside with the Kohler glacier feeding the Dotson ice shelf. A pinning point on the ice shelf is marked with a circle and PP. The background image is the Radarsat Antarctic Mapping Project mosaic which has been overlaid with surface elevation contour lines (blue), the ASAID grounding line (black, Bentley et al., 2014) and the CryoSat-2 Swath coverage in the period from 2010-2016 with colors representing how many years a pixel. (Matsuoka et al., 2021).

important to take the spatial variability into consideration, while keeping in mind that there is a limit to the spatial resolution if

one wants to maximize the signal-to-noise ratio.

Furthermore, high-resolution products come with challenges regarding data volume and availability/accessibility. Both the TanDEM-X, WorldView, and GeoEye data are not directly publicly available, which puts a major limitation on the accessibility. Also, transforming the raw satellite imagery into digital surface models is tedious and may serve as limit of the temporal coverage of a study. In this study, we exploit the Reference Elevation Model of Antarctica (REMA Howat et al., 2019) as

an alternative. REMA provides 2- or 8-m resolution digital surface model strips generated from satellite imagery from the WorldView and GeoEye satellites from 2011-2017. In contrast to the raw satellite imagery, REMA is publicly available, thereby providing opportunities for researchers without direct access to the underlying data.

Chartrand and Howat (2020) have shown that REMA in combination with ICESat and IceBridge can be used to derive basal melt rates and study channel evolution on the Getz Ice Shelf. However, it is also evident that using REMA to derive high spatial

and temporal resolution basal melt rates introduces a new set of problems, in particular the co-registration of the individual digital surface strips and the Lagrangian ice parcel tracking. First, co-registering the REMA digital surface model strips, and transforming them into DEM strips requires several processing steps. Absolute elevation data from e.g. altimetry is needed to correct the relative REMA elevation data (Berger et al., 2017; Shean et al., 2019), but the REMA data is from a period at the very end of the ICESat mission (2003-2009) and before the ICESat-2 launch in 2018. In between, only Operation IceBridge

and CryoSat-2 are available for co-registration. Operation IceBridge carries a laser altimeter among other instruments and was initialized to fill the gap between ICESat and ICESat-2, but at a drastically reduced spatial and temporal coverage. CryoSat-2,





on the other hand, carries a radar altimeter which allows elevation measurements even under cloudy conditions which makes it suitable as a reference for co-registration. Chartrand and Howat (2020) used ICESat for the oldest REMA strips and Operation IceBridge otherwise, which limits the number of REMA strips that can be processed. Second, the Lagrangian thinning and melt
rates rely on co-registring the DEMs, either by feature tracking between two DEMs (i.e., Berger et al., 2017) or by displacing the DEMs using an existing velocity field (i.e., Moholdt et al., 2014). Both methods come with errors which will propagate into the resulting thinning and basal melt. The accuracy of the displacement thereby also influences the highest possible spatial resolution and signal-to-noise ratio.

In this study we use the REMA strips in combination with CryoSat-2 measurements to obtain high spatial and temporal
resolution thinning and basal melt rates of the Dotson Ice Shelf in the period from austral summer 2010/11 to 2017/18. We present and assess the high-resolution Basal melt rates method Using Rema and Google Earth Engine (BURGEE). BURGEE can be run on the Google Earth Engine (GEE), thereby allowing easy access to the data and fast processing on the GEE cloud computing platform (Gorelick et al., 2017). Furthermore, the use of GEE, REMA and CryoSat-2 allows for easy upscalability. We use Dotson as test site since there already exists a detailed basal melt rate study for comparison (Gourmelen et al., 2017).
To investigate the highest feasible resolution, we will perform a sensitivity study assuming that the highest uncertainties are related to the quality of the Lagrangian displacement. We will, furthermore, compare our results to the basal melt rates of Gourmelen et al. (2017) and discuss the different features we observe and the possible influence of a pinning point on basal channel formation and melt rates.

## 2 Theory

The basal mass balance and elevation change of an ice shelf can be observed in both a Eulerian and Lagrangian framework. The Eulerian framework is fixed in space and provides information about the basal mass balance or elevation change at a given point in space. The Lagrangian framework, on the other hand, follows a given ice parcel and assess the basal mass balance or elevation change of that parcel between two places in time, thereby taking the ice flow into consideration. In both cases the basal mass balance can be calculated through a mass conservation approach which in a Lagrangian framework can be expressed
as:

$$\frac{\mathrm{D}H}{\mathrm{D}t} = -H(\nabla \cdot \boldsymbol{u}) + \dot{M}_s - \dot{M}_b,$$
(1)

where $\frac{\mathrm{D}H}{\mathrm{D}t}$ is the Lagrangian ice thickness change, $H$ is the ice thickness, $\nabla \cdot \boldsymbol{u}$ is the divergence of the ice flow, $\dot{M}_s$ is the surface mass balance, and $\dot{M}_b$ is the basal mass balance. By assuming hydrostatic equilibrium, a constant ice density of $\rho_i = 917 \, \mathrm{kg \cdot m^{-3}}$, and a constant sea water density of $\rho_w = 1025 \, \mathrm{kg \cdot m^{-3}}$, the ice thickness can be approximated by

$$H = (h - h_f)\frac{\rho_w}{\rho_w - \rho_i},$$
(2)





where $h$ is the ice shelf surface elevation and $h_f$ the firn air content in meters ice equivalent. Substituting eq. (2) into eq. (1) leads to:

$$\frac{\mathrm{D}h}{\mathrm{D}t} - \frac{\mathrm{D}h_f}{\mathrm{D}t} = (h - h_f)(\nabla \cdot \boldsymbol{u}) + (\dot{M}_s - \dot{M}_b)\left(\frac{\rho_w - \rho_i}{\rho_w}\right), \tag{3}$$

from which we can obtain the basal mass balance:

$$\dot{M}_b = \dot{M}_s - \left(\frac{\mathrm{D}h}{\mathrm{D}t} - \frac{\mathrm{D}h_f}{\mathrm{D}t} + (h - h_f)(\nabla \cdot \boldsymbol{u})\right)\frac{\rho_w}{\rho_w - \rho_i}. \tag{4}$$

## 3 Data

As can be seen from the basal mass balance eq. (4), several auxiliary data sets are required to extract basal melt rates. In this section, we discuss the different data sets used in BURGEE to obtain and evaluate thinning and basal melt rates of the Dotson Ice Shelf.

### 3.1 Surface elevation

To obtain surface elevations of high temporal and spatial resolution we make use of The Reference Elevation Model of Antarctica (REMA, Howat et al., 2019). REMA consists of numerous digital surface model strips of either 2- or 8-meter spatial resolution. They are based on stereo imagery from the WorldView and GeoEye satellites and acquired between 2010 and 2017. The strips are referenced to the WGS84 ellipsoid and are not co-registered. We have chosen to exclude all strips generated using the GeoEye satellites since they suffer from inconsistencies in the surface topography in the form of a striped pattern perpendicular to the satellite flight direction. Besides the strips, a REMA mosaic, made from multiple strips that have been co-registered with CryoSat-2 and ICESat (Howat et al., 2019), will be used as a reference frame to exclude outliers.

To correct the REMA strips for tilt and bias, elevation measurements from the radar altimeter onboard CryoSat-2 are used. CryoSat-2 was launched in 2010 and is the only ice sheet focused altimeter-carrying satellite which has been active throughout our entire study period ranging from austral summer 2010/11 to 2017/18. To transform the waveforms of the CryoSat-2 Level-1B SARIn Baseline-D product to elevations with respect to the WGS84 ellipsoid, we use the leading-edge maximum gradient retracker presented in Nilsson et al. (2016). The downside of using CryoSat-2 is that the radar signals may penetrate into the snowpack, thereby not measuring the direct surface, but some depth into the snowpack. To study this effect, we compared CryoSat-2 measurements to those of the laser altimeter onboard ICESat-2 in the period from 2018 to 2021. From a comparison between neighbouring measurements within 50 m and 5 days, we found a mean penetration depth of -0.4 m with a standard deviation of 2.1 m. We therefore assume that CryoSat-2 elevations can be considered to represent surface elevations.

### 3.2 Surface velocity

Surface velocities are needed to calculate the ice flow divergence in the basal mass balance eq. (4) and to perform a first-order displacement of the DEMs in the chain process of performing the Lagrangian displacement. The surface velocity data is





obtained from the MEaSUREs ITS_LIVE data product (Gardner et al., 2022). These 120 m resolution surface velocities are generated using feature tracking of optical Landsat imagery. Since the velocity field of Dotson has barely changed throughout our study period, we use the 120 m ITS_LIVE composite. Furthermore, we assume that the ice velocity does not vary with depth.

### 3.3 Surface mass balance

Since part of the observed ice shelf thinning may be related to surface processes ($\dot{M}_s$ in eq. (4)) monthly surface mass balance values are obtained from the regional climate model RACMO 2.3p3 (van Wessem et al., 2018). The output from RACMO is given in $\mathrm{mm}$ water equivalent and translated into $\mathrm{m}$ ice equivalent by using an ice density of $917 \ \mathrm{kg \cdot m^{-3}}$. We perform a spatial extrapolation since the 27 km grid does not cover the entire ice shelf. This is done by applying a linear extrapolation over a distance of 5 pixels. Finally, the data is interpolated onto the DEM grid from the original 27 km resolution grid using a

bicubic interpolation.

### 3.4 Firn air content

To obtain the local ice equivalent thickness of the ice shelf, the presence of air in the firn layer needs to be taken into account ($h_f$ in eq. (2)). Estimates of firn air content are obtained from the 27 km resolution IMAU-FDM v1.2A (Veldhuijsen et al., 2022) on a 10-daily basis. The IMAU-FDM is forced with climate data from RACMO which is why they share the same

resolution. This also means that we have applied an identical spatial extrapolation for the firn air content as for the surface mass balance (see Sect. 3.3), to ensure coverage over the entire ice shelf, followed by a bicubic interpolation onto the DEM grid.

### 3.5 Basal melt evaluation

To evaluate BURGEE we compare our results with two existing melt products, based on remote sensing and an ocean model,

respectively. The remote sensing based product is obtained from CryoSat-2 swath measurements resulting in a mean basal melt rate product in the period from 2010-16 at a 500 m resolution (Gourmelen et al., 2017, Fig. 5b). The ocean modeling product (LADDIE) is obtained by a 2D dynamically downscaling of the 3D ocean model MITgcm resulting in basal melt rates at a 500 m resolution (Lambert et al., 2022, Fig. 5c).

To investigate the basal melt pattern we further focus on the thermal forcing and the friction velocity provided by LADDIE,

since basal melt can be approximated by the product of these two terms (e.g., Favier et al., 2019). The thermal forcing is the difference in temperature between the ocean water just below the ice shelf and the freezing point, and can thereby be interpreted as the available heat to melt the ice. The friction velocity, defined as the time-mean ocean velocity just below the ice shelf, describes how effectively the heat is transported to the ice.



## 4   Methods

In the following sections we describe the methodology applied in BURGEE to calculate the basal melt rate using eq. (4). Firstly, the REMA strips have to be transformed to digital elevation models by first accounting for dynamic and static corrections such as tides (see Sect. 4.1) and thereafter through a co-registration to CryoSat-2 (Sect. 4.2). A schematic overview of this procedure can be seen in Fig. 2. The resulting DEMs are then used to obtain both Eulerian (Sect. 4.3) and Lagrangian surface elevation changes (Sect. 4.3), from which the latter is used in the basal mass balance calculation(Sect. 4.5) for which ice flow divergences (Sect. 4.4) are also needed. Finally, a sensitivity study (Sect. 4.6) is performed to assess the highest feasible resolution.

### 4.1   Dynamic and static corrections

Dynamic and static corrections have to be applied to both the REMA strips and the CryoSat-2 elevations to bring all elevations to the same reference frame regardless of geoid, tides etc. However, before that is done all CryoSat-2 and REMA strip elevations that differ more than 30 m from the REMA mosaic are removed as extreme outliers. After this, both the REMA and CryoSat-2 elevation data are referenced to the geoid using the Earth Gravitational Model 2008 (EGM2008, Pavlis et al., 2012).

Due to the underlying ocean beneath the ice shelf, tides ($\Delta h_t$), mean dynamic topography ($\Delta h_{mdt}$), and the inverse barometer effect ($\Delta h_{ibe}$) should also be taken into consideration in the ice shelf elevation corrections. The mean dynamic topography is a static correction like the geoid, for which we use the DTU15MDT which is an updated version of the DTU13MDT (Andersen et al., 2015). Tidal heights are obtained from the CATS2008 model on a 6-hourly interval at a $\sim$3 km spatial resolution. The inverse barometer effect was corrected for by using the 6-hourly NCEP/NCAR sea level pressure reanalysis data (Kalnay et al., 1996), from which residuals were calculated by using a mean sea level pressure of 1013 hPa. The residuals are then scaled by $\sim$0.9948 cm hPa$^{-1}$ to obtain the inverse barometer effect (Wunsch, 1972). Since the oceanic induced corrections are only applicable to the ice shelf itself, the corrected surface elevation is obtained through:

$$h = h_{data} - \Delta h_{geoid} - \alpha \left( \Delta h_t + \Delta h_{mdt} + \Delta h_{ibe} \right), \tag{5}$$

where $h_{data}$ is either the CryoSat-2 or REMA surface elevations, $\Delta h_{geoid}$ is the offset to the geoid and $\alpha$ is a coefficient ensuring a smooth transition from grounded to floating ice as in Shean et al. (2019). This transition is a function of the distance to the grounding line ($l$):

$$\alpha(l) = \begin{cases} 0 & l \leq 0\text{km} \\ \frac{1}{3}l & 0\text{km} < l \leq 3\text{km} \\ 1 & l > 3\text{km} \end{cases} \tag{6}$$

Once the corrections have been applied the elevation data is at the stage marked with an asterisk (*) in Fig. 2.

### 4.2   Co-registration

Since the REMA strips have not been co-registered to the actual surface elevation, strips might be both tilted and vertically misplaced. A co-registration to actual surface elevations therfore is needed. Here, the co-registration of the REMA strips is





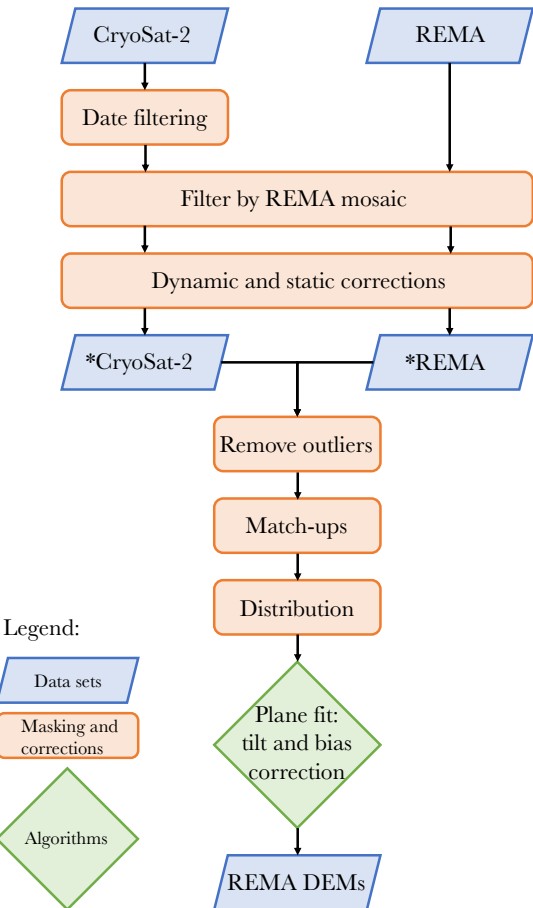

**Figure 2.** Flow chart showing the procedure going from WGS84 referenced CryoSat-2 and REMA digital surface model elevations to fully co-registered DEMs. The dynamic and static corrections are tides, mean dynamic topography, inverse barometer effect, and referencing to the geoid. The asterisk (*) denotes intermediate CryoSat-2 and REMA data sets, before merging the two and applying the two-fold co-registration as described in Sect. 4.2. One being with respect to the CryoSat-2 elevations, and the other with respect to overlapping strips from the same period.

performed by using two consecutive plane-fit co-registration approaches: first with respect to the CryoSat-2 measurements and a second with respect to overlapping REMA strips. The double co-registration is performed to improve the quality, as there might still be small offsets between strips where they overlap. The co-registration to Cryosat-2 is done by correcting for tilt and vertical bias after fitting a plane though the residuals between CryoSat-2 and the individual REMA strips.

Before the co-registration of a REMA-strip to CryoSat-2 can be performed, we defined four criteria that should be fulfilled in the given order: (i) The CryoSat-2 elevations used to perform the co-registration have to be within three months of the





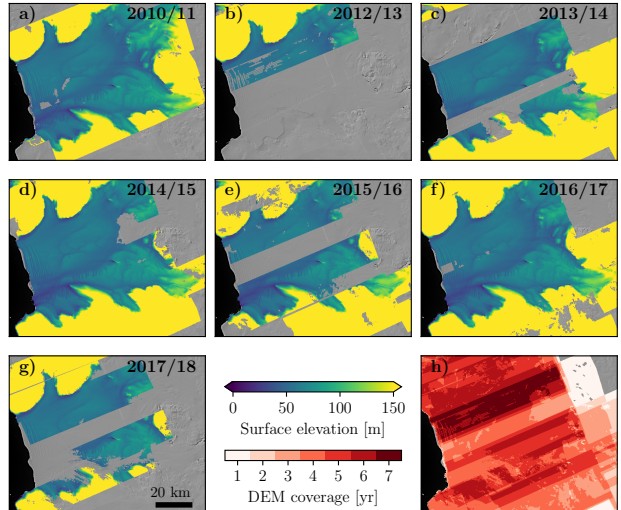

**Figure 3.** Yearly co-registered DEM mosaics ranging from 2010/11 (a) to 2017/18 (g). Notice that there is no data from 2011/12. **h)** Heat map showing the total DEM coverage.

acquisition date of the REMA strip to ensure that the CryoSat-2 elevations are representative for the elevations when the strip

was acquired; (ii) All CryoSat-2 elevations that differ more than 15 m from the REMA strip elevations are excluded, as they are assumed to be outliers; (iii) For each and every REMA strip the number of available CryoSat-2 measurements/match-ups that have fulfilled criterion 1 and 2 should be at least 80 to ensure that we perform a representative plane-fit; (iv) The northern- and southernmost CryoSat-2 points should be at least 40 km apart. Likewise, the CryoSat-2 points furthest separated in the longitudinal direction should be at least 10 km apart. This fourth criterion ensures that the CryoSat-2 measurements are evenly

distributed over the REMA strip.

   The second co-registration is performed on a yearly basis by co-registering all DEMs that overlap at least 25% with the yearly median DEM. The residuals between the DEM strips and the median DEM are thus used to perform the plane-fit. The second co-registration is thereby not applied to all strips.

   The co-registered strips range from austral summer 2010/11 to 2017/18 and have a yearly coverage of up to 98% in 2016/17

(see Fig. 3).

### 4.3   Surface elevation change

To calculate the basal mass balance, the Lagrangian elevation change ($\frac{Dh}{Dt}$ in eq. (4)), is needed. We also assess the elevation change in a Eulerian framework since this provides information about where the ice shelf is thinning and thickening.

   Both the Eulerian and Lagrangian elevation changes are calculated on a tri-yearly basis and as a linear trend throughout the

study period. In both frameworks, July 1st is defined as the start of the year, putting the austral summer in the middle of a year,





with the first year of our study period being 2010/11 and the last year 2017/18. The Eulerian elevation change is thus calculated from the co-registered DEMs and is further cleaned from possible outliers by removing points with a surface elevation change rate of more than $\pm 15$ m yr$^{-1}$.

For the Lagrangian elevation change, the time span over which the elevation change is calculated will be referred to as $\Delta t$.
This means that

$$t_{end} = t_{start} + \Delta t, \tag{7}$$

where $t_{start}$ is the acquisition date of the REMA strip to be displaced and $t_{end}$ is January 1st of the year we wish to displace the strip to.

The most common approaches to assess the Lagrangian elevation change are by displacing the DEMs using a velocity field
(i.e., Moholdt et al., 2014) or by applying a feature tracking algorithm (i.e., Berger et al., 2017). The first approach requires velocities of high quality and resolution and puts strong restrictions on the spatial resolution of the final product. It is, however, computationally efficient, whereas the latter approach is computationally heavy but allows for a higher output spatial resolution. In BURGEE we want to keep the spatial resolution high, while keeping the computational cost low to allow for future study region upscaling. To do so, we first apply a velocity displacement, after which we perform a final correction through feature
tracking. This decreases the search window needed in the feature tracking process, thus reducing the computational time.

The initial displacement of the DEMs is performed by using the ITS_LIVE surface velocities, where all DEMs are displaced to $t_{end}$. This means that all DEMs are roughly aligned to where they would be located at $t_{end}$. However, surface features such as crevasses may not be perfectly aligned for which the second displacement is needed.

This second and final displacement is performed by using the build-in feature tracking algorithm *displacement* in GEE,
which uses orientation correlation. Firstly, yearly DEM mosaics are created using the *quality mosaic function* in GEE to reduce the computational requirements of the feature tracking algorithm. In this displacement, all DEM mosaics are referenced to that of $t_{end}$, thereby aligning all surface features to their position in the DEM mosaic from $t_{end}$. The build-in feature tracking algorithm on the GEE takes in three adjustable parameters: patch width, max offset, and stiffness, which were set to 100 m, 300 m and 3, respectively. The patch width defines the size of the patches/regions to search for within the distance given by
max offset, whereas the stiffness parameter defines how much distortion/warping is allowed. Since ice features may very well change in shape, a lower less rigid stiffness parameter is desired.

The Lagrangian elevation change using the displaced DEM mosaics are then obtained similar to the Eulerian elevation change by applying a linear fit to the DEM mosaics and their corresponding time stamp, which in this case may vary from pixel to pixel. The same outlier criterion of 15 m yr$^{-1}$ is used to mask out the remaining possible outliers.

**4.4 Divergence of the velocity field**

Berger et al. (2017) illustrated the impact different methods of calculating the gradients have on the resulting divergence. They show that traditional methods, i.e., forward, backward, or central differences, increase the signal-to-noise ratio when applied to noisy data such as velocity fields. A way to go around this issue, is to apply a smoothing to the velocity field. However,





this implies an undesired smoothing of steep gradients in i.e., shear zones. Instead, one can use regularized divergences, which
suppresses noise while keeping the data signal (Chartrand, 2011).

In this study we have chosen to use total-variation regularization (Chartrand, 2017), which is an updated version of the
regularization method presented in Chartrand (2011) made especially for multidimensional data, to compute the gradients of
the 120 m resolution ITS_LIVE velocity field. Due to our previous assumption of a constant velocity field in time, we also
assume that the resulting divergence field of Dotson is constant in time. Finally, the divergence field is linearly interpolated
onto the DEM grid.

## 4.5    Basal mass balance

As for the Eulerian and Lagrangian elevation changes, the basal mass balance is also calculated on a tri-yearly basis and as
a trend over the entire study period by using eq. (4). The densities $\rho_i$ and $\rho_w$ are assumed to be constant and do therefore
not depend on $\Delta t$. The surface mass balance, $\dot{M}_s$, is the average yearly surface mass balance from the time of the first DEM
mosaic to the time of the last DEM mosaic measured in meters of ice equivalent. The Lagrangian elevation change, $\frac{\mathrm{D}h}{\mathrm{D}t}$, is
obtained as described in Sect. 4.3. The term $(h - h_f)(\nabla \cdot \boldsymbol{u})$ is calculated on a yearly basis regardless of the chosen $\Delta t$. If
$\Delta t > 1\mathrm{yr}$ the average of the years considered is taken. The divergence field and the firn air content are added as bands to the
DEMs before applying the velocity displacement. This is done to ensure that the firn air content and divergence field in each
pixel corresponds to that of the surface elevation used in the quality mosaics. The firn air content and divergence field are
therefore also displaced, and their values correspond to those at the original position and time (in the case of firn air content)
of the DEM strips. Hence, when $\Delta t$ is the entire study period, all DEM mosaics are taken into consideration when calculating
$(h - h_f)(\nabla \cdot \boldsymbol{u})$, and individual ice parcels can be considered to be tracked on a yearly basis. This method takes the entire path
which an ice parcel goes through from 2010/11 to 2017/18 into consideration when calculating the basal mass balance over the
entire study period. Finally, the change of firn air content with time, $\frac{\mathrm{D}h_f}{\mathrm{D}t}$, is obtained by taking the linear trend of the firn air
content band of the DEM mosaics over the period $\Delta t$.

## 4.6    Sensitivity experiment

Since the signal-to-noise ratio of basal mass balance is expected to increase when increasing the spatial resolution, we per-
formed a synthetic experiment to assess the impact of spatial resolution on the basal melt rate in BURGEE. The sensitivity
experiment is based on the assumption that the Lagrangian displacement is one of the main contributors to the basal melt rate
uncertainty. Within this experiment we used the perfectly aligned annual DEM mosaics from 2010/11 and 2014/15, since they
have a high coverage and are relatively far apart in time thus putting stronger requirements on the displacement algorithm
applied. Based on both aligned annual DEM mosaics, two different basal mass balance maps are obtained using two different
Lagrangian elevation changes. This is first done from the perfectly aligned DEM mosaics resulting in the *true* basal melt rate.
Second, the otherwise perfectly aligned 2010/11 DEM mosaic is displaced based on the four year accumulated ITS_LIVE
error fields to create an alternative aligned DEM mosaic that incorporates the displacement uncertainty due to velocity errors.
The Lagrangian elevation change is then calculated using the error-displaced 2010/11 DEM mosaic and the before mentioned



2014/15 DEM mosaic. The difference between the resulting basal melt rates following from the two Lagrangian elevation changes is then calculated at various resolutions to see what resolution is required for the artificial error to cancel out.

# 5 Results

## 5.1 Evaluation of BURGEE results

The Eulerian and Lagrangian surface elevation trends over the entire study period (2010/11-2017/18) are shown in Fig. 4 at a 50 m resolution. Owing to its along-flow coordinate system, the Lagrangian elevation change has a much smoother pattern compared to the Eulerian framework. In both frameworks, the channel described by Gourmelen et al. (2017) (red arrow) shows pronounced thinning. The Eulerian elevation change shows elevation decrease along and west of the channel, whereas the area east of the channel shows a more irregular pattern. Similarly, we see almost no elevation change east of the channel in the Lagrangian elevation change, and a smoother pattern due to the along-flow framework. Three areas stand out with high Lagrangian elevation changes, namely the channel (red arrow), at the border towards the Crosson Ice Shelf (orange arrow), and the grounding zone at the inflow of the Kohler glacier (green arrow). The latter is not fully resolved in Gourmelen et al. (2017), presumably due to limited CryoSat-2 coverage (Fig. 1). Furthermore, there is a smaller area towards the Crosson ice shelf with a high Lagrangian elevation change (orange arrow).

The basal melt rate at a 50 m resolution over the entire study period can be seen in Fig. 5a, which shows a similar pattern as the Lagrangian elevation change. Figure 5d-h show the 3-yearly basal melt rate trends also at a 50 m resolution. There is a clear spatial consistency throughout the entire period, though melt rate magnitudes do seem to show temporal variability. A striped pattern is visible on some of the 3-yearly maps (e.g 2011/12-2014/15 and 2012/13-2015/16) due to the DEM mosaic coverage seen in Fig. 3. The varying coverage implies that the trend is taken over different time periods whenever there is missing data in the DEM mosaics. Furthermore, the Lagrangian displacement is performed with respect to the latest DEM mosaic (Sect. 4.3), which is what is causing the higher melt rates in the center of Dotson in the 2012/13-2015/16 product. Focusing on the basal melt rate of the entire study period (Fig. 5a) we see that high melt rates are present along the channel (red arrow), near the Kohler grounding zone (green arrow), and towards the Crosson Ice Shelf (orange arrow). These are all features that are present in both Gourmelen et al. (2017) (Fig. 5b) and Lambert et al. (2022) (Fig. 5c). Likewise, the overall pattern is similar in all three products. In our study, we also see a channelized pattern running along the western ice shelf margin (blue arrow). The narrow coastal channel along the western ice shelf margin (blue arrow) is not evident in the results of Gourmelen et al. (2017), it is, however, present in Lambert et al. (2022). Figure 6 shows a zoom-in of the channel, from which it becomes apparent that our melt rate product shares a lot of similarities with those of Lambert et al. (2022), especially when it comes to features of smaller scales such as the coastal channel. The striped pattern in the northeastern part of the main channel in our product is due to the varying and often limited coverage of the REMA strips (Fig. 3h).

The main channel (Fig. 5a, red arrow) corresponds to the same channel observed by Gourmelen et al. (2017) (Fig. 5b) and Lambert et al. (2022) (Fig. 5c). Its pattern, bending towards the margin of the ice shelf, is explained by buoyancy forcing of the low-salinity melt water plume (Lazeroms et al., 2018). In all three studies, a convergence zone, causing a narrow sharp plume



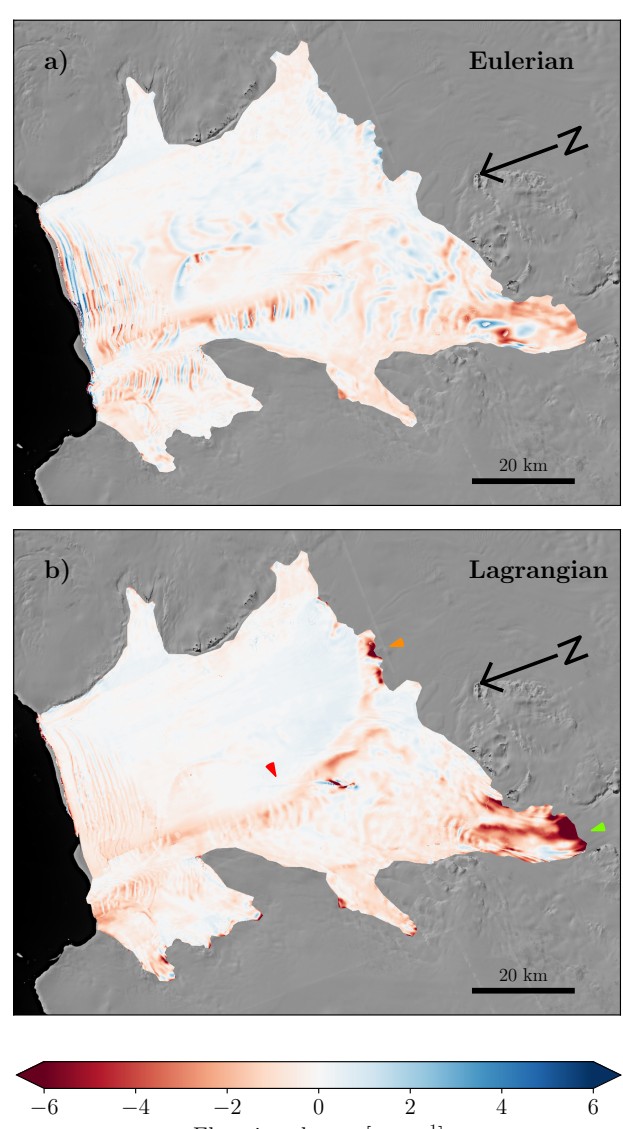

**Figure 4.** Eulerian (a) and Lagrangian (b) surface elevation trends from 2010/11-2017/18 at a 50 m resolution. The red arrow marks the main channel, the green arrow marks the Kohler grounding zone, and the orange arrow marks the high elevation changes towards the Crosson Ice Shelf.

with high melt rates, is found just east of the pinning point (Fig. 5a, cyan arrow). The positioning of this convergence zone may be determined by the pinning point, since melt water plumes causing channel formation have a tendency to align along western topographic boundaries in this area (Lambert et al., 2022).





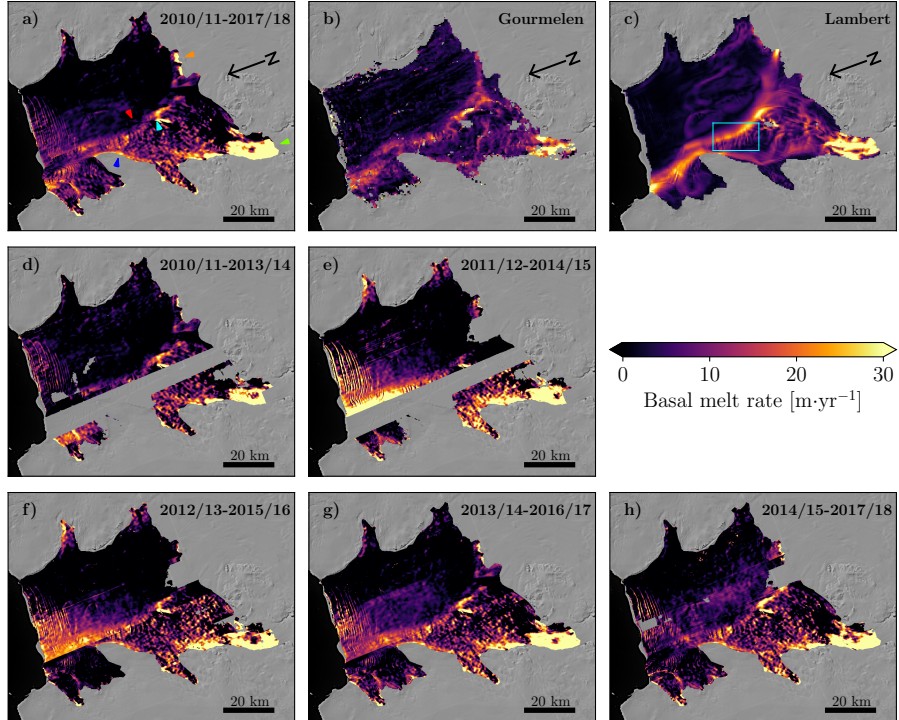

**Figure 5. a)**: Basal melt rate trend from 2010/11 to 2017/18. The arrows point out the main channel (red), the smaller channel along the western ice shelf margin (blue), the pinning point below the ice shelf (cyan), the Kohler grounding zone (green), and the high melt rates near the Crosson Ice Shelf. **b)**: Basal melt rate from Gourmelen et al. (2017). **c)**: Basal melt rate from Lambert et al. (2022) with the box marking the zoom-in in Fig. 9. **d)**-**h)**: Tri-yearly basal melt rate trends.

## 5.2 Results from the sensitivity experiment

Figure 7 shows the result of the sensitivity study where we created an alternative aligned DEM mosaic by displacing the
correctly aligned DEM based on the error of the ITS_LIVE velocities under the assumption that the quality of the Lagrangian displacement is the main contributor to the basal mass balance uncertainties. In the part of the ice shelf with fewer surface undulations (10-15 km), the errors resulting from the artificial displacement cancel out to a large degree already at a 50 m resolution. In areas with stronger surface undulations, it requires a coarser resolution for the error to cancel out. However, it should be noted that even at a high resolution of 50 m, the resulting error of the cross section is within $\pm 2\,\mathrm{m\,yr^{-1}}$. Furthermore,
the highest errors correlate with high melt rates. Based on these findings, we have chosen to offer our product at a 50 m and a 250 m resolution (https://doi.org/10.4121/21841284). It should be noted, however, that the Lagrangian displacement is not the only error source, since all data and assumptions used to calculate the basal mass balance (eq. (4)) come with errors and uncertainties, which is why these numbers cannot be considered as true uncertainties of the final product.



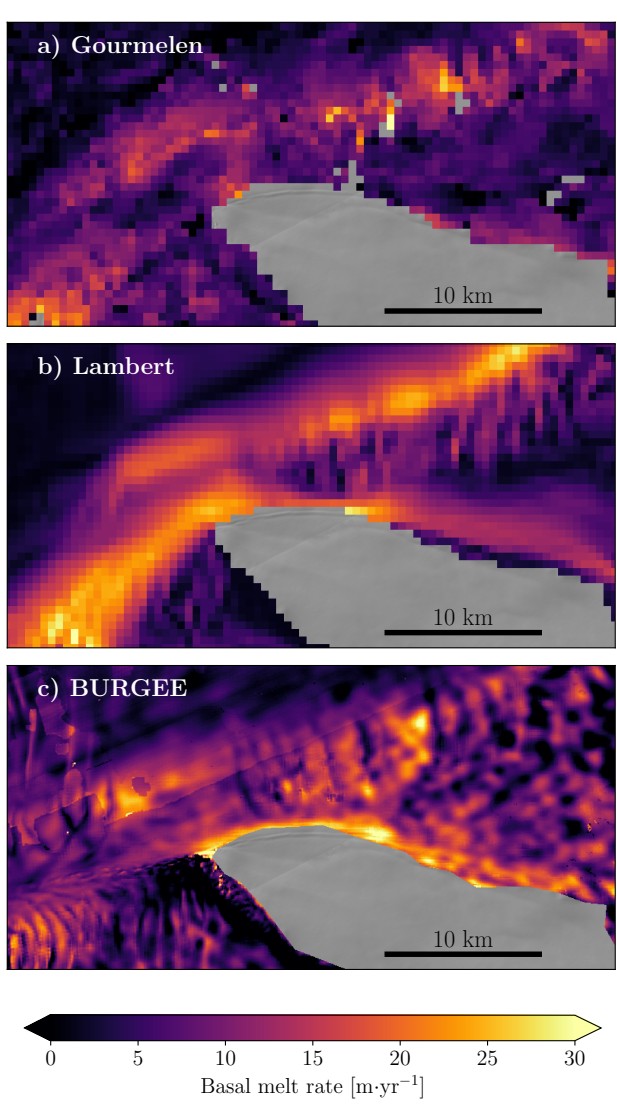

**Figure 6.** Zoom-in of basal melt rates of the western ice shelf margin from **a)** Gourmelen et al. (2017), **b)** Lambert et al. (2022) and **c)** this study.



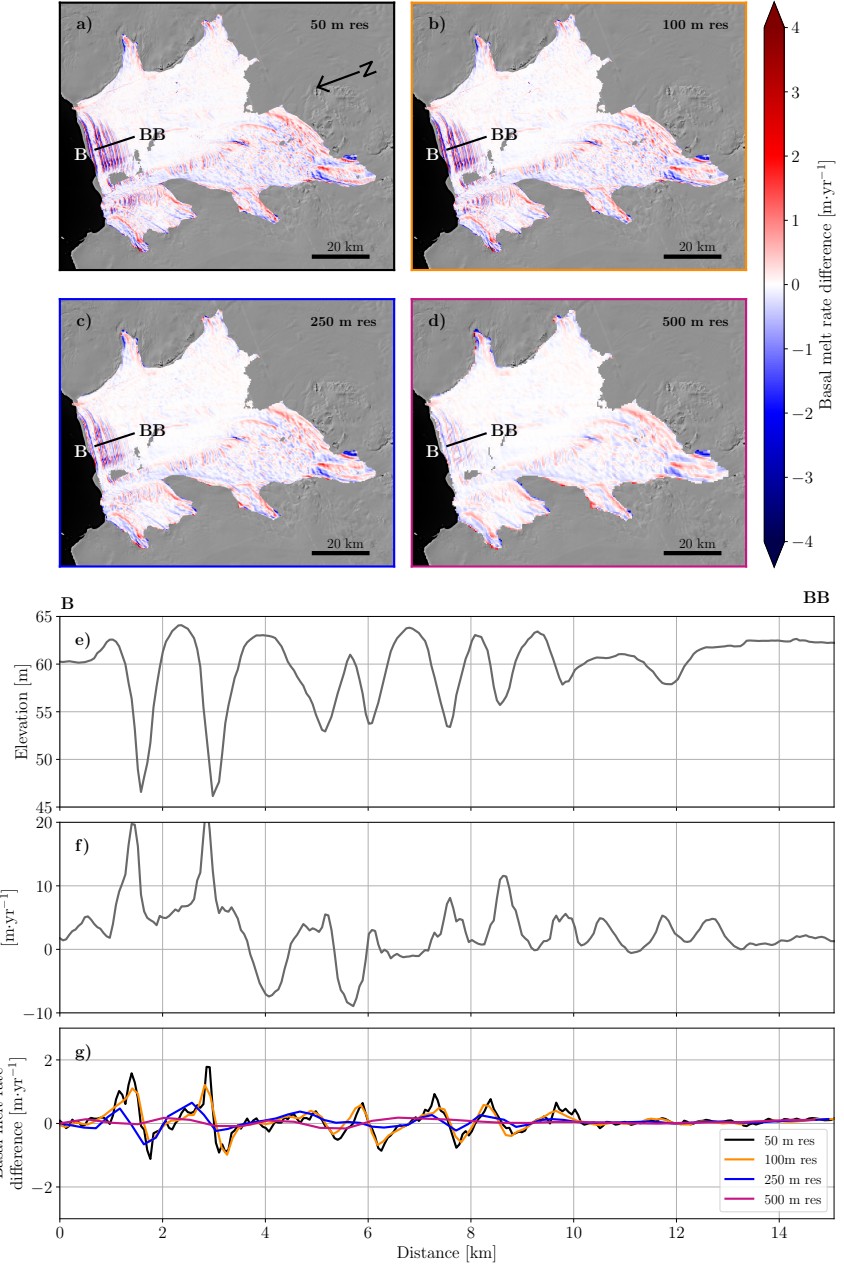

**Figure 7. a)-d)** shows the difference between the basal melt rate obtained from the correct and the erroneous Lagrangian elevation changes at 50 m (a), 100 m (b), 250 m (c), and 500 m (d) resolutions. **e)** shows the perfectly aligned 2010/11 DEM mosaic at the B-BB cross section. **f)** shows the basal melt rate obtained from the correct DEMs at the B-BB cross section. **g)** shows the basal melt rate differences at the B-BB cross section at 50 m, 100 m, 250 m, and 500 m resolutions. Note that f) and g) use a different range on the y-axis.

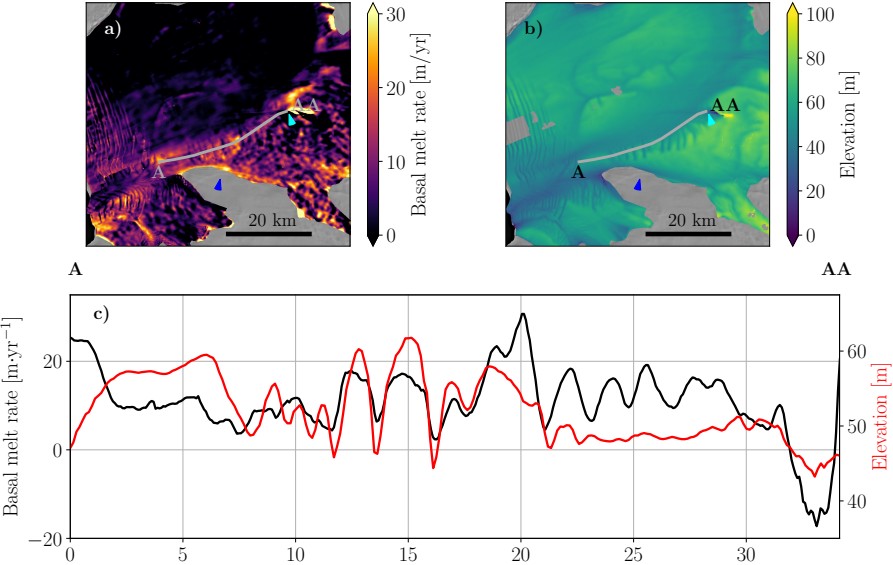

**Figure 8. a**) Basal melt rate from 2010/11-2017/18; **b**) 2016/17 DEM, with the gray line marking the cross section A to AA in c) and d); **c**) Basal melt rate at the cross section marked in a) and b); **d**) Surface elevation from the 2016/17 DEM, prior to any Lagrangian displacement, at the cross section marked in a) and b). The distance in c) and d) is with respect to the left end-point (A) of the gray line in a) and b).

## 5.3 Interpretation of the melt pattern

In the vicinity of the pinning point (5a, cyan arrow), the main channel shows a smooth pattern which changes to a more wavey pattern downstream. Roberts et al. (2018) hypothesize that such a wavey pattern can originate near pinning points due to ocean heat variability, where periods of increased available ocean heat induces enhanced thinning and a reduced back-stress over the pinning point and vice versa. Through convergence and divergence, the resultant temporal variability in ice speed translates into an alternating pattern of thick and thin ice. To investigate the influence of the pinning point on the spatial melt variability within the main channel, we have focused on a transect following the basal channel from the pinning point to the ice shelf front (Fig. 8, transect A to AA and pinning point marked with the cyan arrow). We can see clear surface undulations along the channel that emerge downstream from the pinning point. We also notice higher melt rates coinciding with these surface undulations, especially around 12-22 km, where we see that higher melt rates align with higher surface elevations, implying greater ice thicknesses, and vice versa.

To investigate the relation between the surface undulations and the basal melt rate, we disentangle basal melt into its two major components: thermal forcing and friction velocity. The thermal forcing determines the locally available heat for melting, whilst the friction velocity determines the efficiency of turbulent heat exchange toward the ice shelf base. The thermal forcing and friction velocity, simulated by LADDIE, are shown in Fig. 9, alongside with the LADDIE basal melt rates and draft. Fig. 9e-f further shows the friction velocity and the thermal forcing as a function of the melt rate, respectively. It is evident that the



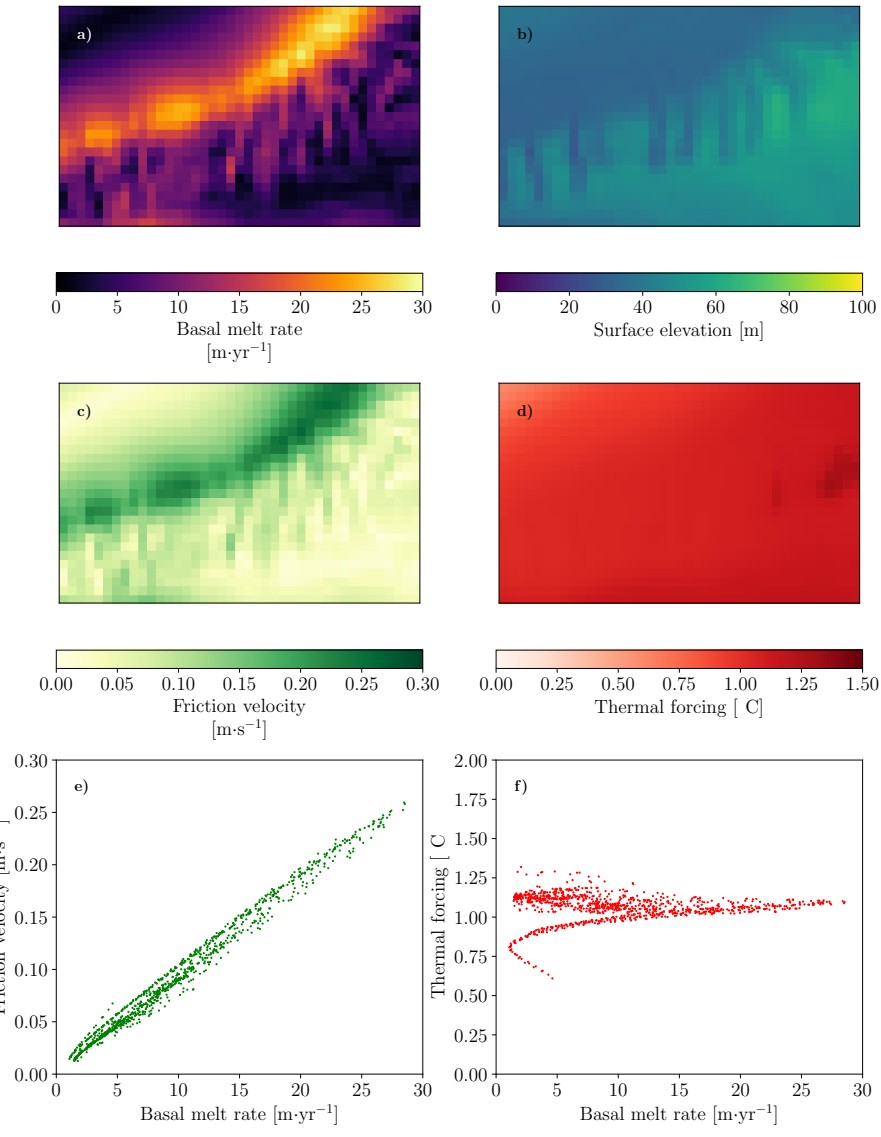

**Figure 9.** Model outputs from LADDIE within the box marked in Fig. 5c: **a)** Basal melt, **b)** surface elevation, **c)** friction velocity, **d)** thermal forcing, **e)** friction velocity as a function of basal melt, and **f)** thermal forcing as a function of basal melt.

friction velocity has a high correlation with the melt rate, implying that it is the main driver of spatial melt variability at fine scales. The friction velocity is affected by the undulations. Where the melt water plume encounters thick ice, it is squeezed vertically, leading to convergence and a local acceleration. This locally enhanced friction velocity increases the heat transfer and consequently the basal melt. As basal melt is typically highest in regions of thick ice (Fig. 8), this interaction between ice





topography and melt is a negative feedback that should smoothen out the undulations downstream. This negative feedback may
explain the weakening signature of the undulations towards the ice shelf front.

## 6  Discussion

We show how REMA in combination with CryoSat-2 is capable of obtaining high resolution basal melt rates of the Dotson
Ice Shelf. The use of CryoSat-2 in BURGEE allows to process significantly more REMA strips than when using ICESat and
Operation IceBridge only as in (Chartrand and Howat, 2020). We observe the same large scale melt pattern consisting of
one larger melt channel as previous studies (Gourmelen et al., 2017; Lambert et al., 2022). Additionally, BURGEE allows to
observe small-scale melt features which would go unnoticed in lower-resolution altimetry-based products such as Gourmelen
et al. (2017) and Adusumilli et al. (2020). This especially becomes evident in two different cases.

Firstly, our elevation maps reveal that surface undulations appear after a pinning point on the ice shelf. Here, we also find a
correlation between melt rates and ice thickness along the main basal channel. The link between surface undulations and basal
melt rates is further supported by Watkins et al. (2021) who found a clear relationship between pinning points and roughness,
and a correlation between the latter and basal melt. Furthermore, modeling studies suggest shear zones and topographic features
to have a possible impact on basal channels and their formation (Gladish et al., 2012; Sergienko, 2013). Also, Roberts et al.
(2018) suggest that varying ocean temperatures leads to ice shelf thickness change and thereby a change in the back-stress
over ice rumples causing a wavey pattern as the one we observe being initialized at the pinning point. Model output from
LADDIE suggests that the friction velocity is the driver of the increased melting in regions of greater ice thickness, due to
compression and divergence of the melt plume. Eastward, a convergence zone indicates that the channel pathway may also
be influenced by the pinning point. Melt water plumes causing channel formation have a tendency of occurring at western
topographic boundaries in this area of Antarctica (Lambert et al., 2022) which we for instance also observe with the coastal
channel. The eastward convergence zone suggests that the pinning point is big enough to act as a western topographic feature.
Without the pinning point the melt water plumes may thus converge elsewhere. We therefore postulate that the pinning point
impacts the spatial variability within the channel and possibly also the channel pathway due to the convergence zone just east
of the pinning point.

Secondly, we observe a smaller channel running along the western margin of the ice shelf which is also present in LADDIE
(Lambert et al., 2022). We note that hydrostatic equilibrium is assumed in our approach and has only been addressed through
our gradual correction for tides, mean dynamic topography, and inverse barometer effect (eq. (6)), which may introduce ar-
tifacts where equilibrium is not reached. The grounding zone product obtained from ICESat-2 (Li et al., 2022) indicates that
hydrostatic equilibrium is reached within a few kilometers from the grounding line for most of the ice shelf, except for en-
closed bays narrower than $\sim 10\,\mathrm{km}$. In addition, the results of LADDIE rely on the provided ice shelf topography (Morlighem
et al., 2020) which is similarly based on the assumption of hydrostatic equilibrium as well as smoothing near the grounding
line. Since the small channel which we observe is not located in an enclosed bay and Coriolis deflection is expected to form a
western boundary current here, we are confident that the channel is not an artifact of the hydrostatic equilibrium assumption.





Finally, the spatial coverage is improved when using a combination of REMA and CryoSat-2. CryoSat-2 on its own cannot cover all parts of an ice shelf, especially those surrounded by topographic features such as mountains. Furthermore, we can assess temporal changes on a 3-yearly basis, though it should be noted that a shorter time frame puts constrains on the spatial coverage due to the yearly REMA coverage. Using elevation data from the CryoSat-2 swath mode alone (Gourmelen et al., 2017) would likely yield a higher temporal resolution, but would not resolve the small-scale features which we can capture with BURGEE. Adusumilli et al. (2020) used a wide range of remotely sensed surface elevations to obtain basal melt rates at quarter-yearly resolution but at the cost of the spatial resolution. Dotson Ice Shelf did not show noticeable changes in basal melt over the study period, however, ocean observations from the Dotson ice shelf cavity do show variations on seasonal timescales of inflow of warm circumpolar deep water (Jenkins et al., 2018; Yang et al., 2022). This could imply seasonal variability in the basal melt rates, as has been observed at both the Nivlisen Ice Shelf in East Antarctica (Lindbäck et al., 2019) and the Filchner-Ronne Ice Shelf (Vaňková and Nicholls, 2022). To investigate seasonal changes using BURGEE, more DEMs than currently available from REMA would be needed. If REMA-2 or similar data products based on future missions were to provide such a higher temporal coverage, studying seasonal or interannual variations in basal melt would become in reach with BURGEE.

The clear advantage of BURGEE is the high spatial resolution and the use of Google Earth Engine. The latter allows one to efficiently and fast process large amounts of data, while the build-in data catalogue drastically reduces the amount of data which has to be downloaded locally. Furthermore, the choice for no site-specific tuning makes that the methodology can be easily applied to other ice shelves in Antarctica. By incorporating upcoming and more up-to-date elevation datasets, we might then be able to assess basal melting at 50 m resolution and at greater temporal resolution than three years. By doing so to other ice shelves influenced by pinning points our product may also help answering what the effect of those are on the basal melt pattern. But more importantly, it would help us to assess the (in)stability of other ice shelves and locating weak spots on them.

## 7 Conclusions

In this study we have showed that the Reference Elevation Model of Antarctica can be used to obtain high-resolution surface elevation changes and basal melt rates of the Dotson Ice Shelf. We perform a sensitivity study which further supports the trustworthiness of the observed small-scale features and indicates that a 50 m spatial resolution is feasible. BURGEE reveals features, such as a smaller melt channel, which has not yet been observed by remote sensing products of coarser resolution. Furthermore, we find strong indications that a pinning point on the ice shelf influences the spatial melt variability within the main channel and that it may be controlling the position of of a warmer ocean plume, thereby impacting the pathway of the main melt channel. Overall, the found small-scale features underlines the importance of high-resolution basal mass balance products as our product can help future studies to provide answers to the causes behind them.

Finally, BURGEE contains no site-specific tuning, which means that it can easily be applied to other ice shelves. Of course, our assumption of a constant velocity field in time does not hold on all ice shelves and should thus be adjusted. Nonetheless, with the right computing sources, this study could be up-scaled to a Pan-Antarctic study of high-resolution basal mass balances.



*Code and data availability.* The BURGEE code is available on github (https://github.com/aszinck/BURGEE). Both elevation and velocity
data is publicly available: REMA (https://www.pgc.umn.edu/data/rema/), CryoSat-2 (https://earth.esa.int/eogateway/missions/cryosat/data),
and ITS_LIVE (https://its-live.jpl.nasa.gov/). Derived surface elevation changes and basal melt rates are available from https://doi.org/10.
4121/21841284.

*Author contributions.* The research was designed by A.P. Zinck, B. Wouters, and S. Lhermitte, and carried out by A.P. Zinck with inputs
from B. Wouters and S. Lhermitte. E. Lambert made significant contributions to the interpretation of the results. All authors helped with the
scoping of the paper which was lead by A.P. Zinck with inputs from all authors.

*Competing interests.* B. Wouters and S. Lhermitte are members of the editorial board of The Cryosphere.

*Acknowledgements.* This study is a part of the HiRISE project which is funded by the Dutch Research Council (NWO, no. OCENW.GROOT.2019.091).
The authors thank N. Gourmelen for providing basal melt rates of the Dotson Ice Shelf from remote sensing.



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
