# Peer review of "REMA reveals spatial variability within the Dotson Melt Channel"

_The Cryosphere, 2023_

## Referee Comment (RC1)

Review for "REMA reveals spatial variability within the Dotson Melt Channel"

The authors use a novel cloud-based method to prepare DEMs and perform basal melt rate calculations called BURGEE (Basal melt rates Using REMA and Google Earth Engine). They show that this method produces high-resolution (50 m) gridded melt rate results that match closely with two other lower-resolution (500 m) studies for large ice shelf features, but that the higher-resolution product captures smaller melt features as well. They also show that errors in basal melt rates due to shifting DEMs using surface velocity maps are reduced at lower resolutions (250 m), without significant loss of spatial variability. Finally, they put forward explanations for some of the spatial variability in surface elevation and melt rates in the vicinity of two basal channels.

This paper fits the topic of the journal and presents a novel, scalable, open-source method for producing basal melt rate maps from REMA DEMs. Overall, I find this study to be well-thought-out, and the figures are *excellent* in both quality and content. However, I have several concerns about the organization and clarity of the paper. I have some recommendations to address these concerns:

1. Several of the thresholds within the steps used to register and correct the REMA strips seem arbitrary to me (Section 4.2) – I would like to see brief justifications for them:
   - L. 200 (criterion ii) Why is the 15 m CS-2 vs REMA difference used to filter CS-2 outliers? I see how this filter would remove control points where either CS-2 or REMA is erroneous, but if the strip is consistently 15+ m too high/low but otherwise fine, it will be excluded. If the authors wish to remove erroneous CS-2 points before using them for registration, I recommend filtering CS-2 using published quality flags and/or by comparing CS-2 to the published REMA mosaic and/or ICESat-2 and removing outliers that fall outside some range around a spatially-averaged elevation or elevation difference (for example, interpolate REMA mosaic to CS-2 points, take moving mean of both sets of points or their differences over some along-track distance, and remove CS-2 elevations that are above/below 2 standard deviations from the moving mean) – this would likely be reported in Section 3.1.
   - Line 202 (criterion iii) Why is the minimum threshold for the number of match-ups 80? It seems that this minimum should depend on the relative size and resolution of the strip and footprint of CS-2.
   - L 203-205 (criterion iv) Similar to my comment above, it is not clear why the north-south and east-west distances between distal CS-2 points should be >40 and 10 km, respectively. This is effectively setting a minimum size for the strips – and the strips could easily be filtered earlier by area or length and width. I think this step is fine, but please include a specific justification for the minimum distances. It would also be helpful to include a figure showing the CS-2 reference ground tracks, or actual ground tracks used in this study, and how they are oriented compared to the strips (for example, add the reference ground tracks to Fig. 3h), and a discussion of how these orientations informed the thresholds that have been set.

2. The process by which the annual DEM mosaics shown in Fig. 3 are produced is not outlined clearly. I recommend including a separate section in the methods addressing this, separately from the (co-)registration of each strip. This omission also leads to confusion in interpreting the basal melt rate results in Fig. 5, which show maps with nearly complete coverage for combinations of years in which the mosaics are not complete. How are the complete coverage maps in Fig. 5 obtained? Are supplementary strips used that were not included in the mosaics?

3. Related to point 2, the description in Section 4.3 of the Lagrangian displacement method is unclear. I made connections (perhaps mistakenly) between both a. displacing strips acquired within a given austral summer as part of the co-registration process to produce an annual mosaic, and b. displacing annual mosaics to t_end in order to calculate rates of elevation change. However, the distinction between these processes is not clear as written, and I think that devoting a section to describing the mosaicking process will greatly help readability. It may also help to add a panel to Fig. 2 showing the workflow for the mosaicking.

4. I recommend reorganizing section 4.5 for clarity, particularly to distinguish the different time scales used in basal mass balance estimates. Please see major comments 2 and 3 above about how DEM mosaics are created and how DEM strips/mosaics are displaced to estimate Lagrangian changes. I find a number of possible inconsistencies that could be clarified by separating bits from Sections 4.2 and 4.3 to form a new section devoted to describing how the annual mosaics are made:

If we consider an example in which we calculate a tri-yearly basal melt rate between the years 2011/12 – 2014/15:

- "$M_s$ is the average yearly surface mass balance from the time of the first DEM mosaic to the time of the last DEM mosaic" indicates that the surface mass balance is the average value of all annual values between 2011/2012 and 2014/15.
    - However, section 3.3 indicates that monthly SMB values are used – so is the 2011/2012 annual DEM associated with the average of all monthly values between 1 July 2011 – 30 June 2012 OR is each strip associated with $M_s$ most close to its collection time, and the $M_s$ is "mosaicked" similarly as the strips to create an annual $M_s$ map?? Please clarify here and/or in Section 3.3/new mosaicking section.
- "The divergence field and the firn air content are added as bands to the DEMs before applying the velocity displacement" indicates that The 2011/12 annual mosaic and its firn air content and divergence field are shifted to the ice's position in 2014/15.
    - However, Section 3.4 indicates that firn air content is obtained on a 10-daily basis – so is the 2011/2012 annual DEM associated with the average of all 10 day values between 1 July 2011 – 30 June 2012? OR is each strip associated with the firn air content most close to its collection time, and the firn air content is "mosaicked" similarly as the strips to create an annual firn air content map? Please clarify here and/or in Section 3.4/new mosaicking section.

- L. 261 indicates that (h − hf )(∇ · u) is calculated on a yearly basis – so is each annual DEM associated with its own (h − hf )(∇ · u) map? Please clarify here and/or in Section 4.4/new mosaicking section.
  - Related, "If Δt > 1 year, the average of the years considered is taken" – does this mean that for 2011/12 – 2014/15 that the (h − hf )(∇ · u) used in Eq. 1 is the mean of the 2011/12 (h − hf )(∇ · u) map and the 2014/15 (h − hf )(∇ · u) map? Please clarify.
- "The firn air content and divergence fields are therefore also displaced, and their values correspond to those at the original position and time (in the case of firn air content) of the DEM strips" – this indicates that divergence fields and firn air contents are calculated/interpolated to each strip before annual DEM mosaic are created. I recommend this be moved to a new section about how the annual DEM mosaics are created.

- L. 266-269 about the entire study period relates to later questions about how the maps in Fig. 4 are produced. I read that all annual mosaics are shifted to their position in 2017/18, moving ice parcels forward one year at a time, and recalculating (h − hf )(∇ · u) after each 1-year shift – is this right? Please clarify if not. Please also clarify how Mb is calculated over the entire time period – are elevation changes, firn air column averages, and $M_s$ averages computed between every combination of flow-shifted DEMs (i.e. 2010/11 – 2011/12 and 2010/11 – 2012/13... and ...2010/11 – 2017/18, and so on), or between consecutive DEMs (i.e. 2010/11 – 2011/2012 and 2011/12 – 2012/13 and so on), OR between each DEM and the 2017/18 DEM (i.e. 2010/11 – 2017/18 and 2011/12 – 2017/18 and so on)? Is the map reported in Fig. 4 the average or median across all combinations, or something else?

5. Currently, some contextual discussion is present in the Results section, (e.g. how main channel's bend may be caused by buoyancy forcing of a meltwater plume; how variable melting pinning points may cause surface undulations; postulation about how the hydrostatic assumption might obscure the western margin melt channel in other studies), which makes parts of the Discussion seem like a recap of the results. I recommend reorganizing the Results and Discussion sections a bit so that the Results presents only the authors' novel findings, while the discussion places them in context.

6. Please check for consistency when describing the locations of ice shelf features. I recommend explicitly naming the two basal channels discussed to remove ambiguity (for example, abbreviate the large, more central basal channel DMC (Dotson Main Channel) and the smaller channel at the western shear margin DWC (Dotson Western Channel) or similar). I recommend including a brief section on the morphology of the entire ice shelf, either in the Introduction or beginning of the Results (since a major result of this study is high-resolution elevation maps) in which the features can be identified. They could also be marked on this paper's Fig. 1 with superimposed lines of different colors/weights/styles as in Fig. 1 of Dow et al. (2018, Science Advances – Nansen ice shelf channels). This will make it easier for the reader to orient themselves throughout the paper.

**Minor scientific comments:**
- L. 22  I recommend being more specific about what the potential consequences of basal melting and ice shelf thinning may be (e.g. ice shelf/ice sheet mass loss and Sea level rise)
- L. 49-50 Most literature suggests that this loss-of-lock near steep gradients is unique to radar altimetry – I recommend including a citation here
- L. 50  At this point it is not clear why Fig. 1 is included, since there is no indication so far in the introduction that CS-2 data were used in this study, or that this study focuses on Dotson. I recommend moving L. 50 from "From Fig. 1…" to 54 to the paragraph starting in L. 84, including a brief reminder of the limitations of radar altimetry on steep slopes
- L. 55-57 The impact of the resolution is relative to what questions are being answered – I recommend being more specific about what is lost due to varying resolutions across studies/methods. Furthermore, in L. 56 "the chosen temporal and spatial resolution" is a little misleading, because the spatiotemporal resolution and range is constrained by data availability, not just choice – please revise.
- L. 78/L. 354  Chartrand & Howat used OIB for registration where available and CS-2 otherwise (ICESat and REMA do not overlap temporally) – please revise
- L. 83  Please specify what metric (elevation, elevation change, basal melt rate) the signal-to-noise ratio is concerning for
- L. 103 Please specify the signs for melt/accumulation in Eq. 1
- L. 122-131 I recommend including some or all of the detail about how both CS-2 and REMA strips are filtered prior to registration (including moving L. 173-174 to here).
- L. 154/Section 3.5 Consider renaming this "basal melt rate comparison products" or similar since "evaluation" connotes methodology.
- L. 173-174 I recommend moving the sentence beginning "However, before all that is done…" this to Sect. 3.1 and revising surrounding sentences as appropriate
- L. 179-182 Applying the tide and ibe corrections requires knowledge of the time the elevation data were collected – please specify how the collection times of the strips were determined and discuss the implications of using 6-hourly corrections
- L. 186 Please specify which grounding line product was used to calculate the correction transition
- L. 190/Section 4.2 Please see major comments 1 & 2 above
- L. 214 Please provide brief justification for using tri-yearly basis for elevation changes – why not annual?
- L. 217-218, L. 244 Similar to major comment 1 above, please provide brief justification for removing points with elevation change rate of > 15 m/yr as opposed to deviation from a spatial mean. Particularly in Lagrangian, this could obscure fast, small scale processes like rift opening (which may be fine for this study since it is focused on melt rates and not fracture)
- L. 224-241 Please see major comment 3 above
- L. 256-270/Section 4.5 Please see major comment 4 above
- L. 275 Related to major comments 2-4 above, please define what a "perfectly aligned annual DEM mosaic" is. I would recommend avoiding using "perfectly" unless it is specifically defined earlier.
- L. 283 Please include more detail about the various resolutions used and why

- L. 286/Fig. 4 Related to major comment 4 above, please describe how the 2010/11 – 2017/18 trends were calculated
- L. 293 Related to comment on L. 186 above – please describe how the Dotson/Crosson border was defined and/or why was the ASAID grounding line is used to define the Dotson ice shelf
- L. 294-295 isn't the "smaller area" the same as "at the border towards the Crosson" in L. 293? Remove or revise.
- L. 299 Specify which panels of Fig. 5 the striped pattern is visible
- L. 300 varying coverage – see major comment 2
- L. 303 spatial description – see major comment 6
- L. 313 "bending toward the margin" – which margin? See major comment 6
- L. 315 I recommend including a velocity and/or strain rate map in one of the figures so the reader can see the convergence zone clearly
- L. 321-322 see major comment 6
- L. 324 Please specify what the cross section is and what quantity (basal melt, thinning?) has an error of +- 2 m/yr
- L. 330 please specify what quantity (elevation, melt rate?) has a smooth/wavy pattern
- L. 338-339 Please clarify what is meant by "implying greater thicknesses", or consider reframing this as evidence for deeper basal drafts experiencing more melt, which is seen elsewhere in literature
L. 362-363 Alley et al. (2019, Sci. Adv.) also shows marginal channels observationally, and Alley et al., (2016, Nat. Geosc.) identify this western marginal channel (not plotted, but in the shapefiles)
L. 381 It seems that the authors should be able to verify the presence of a surface depression associated with the basal channel from the DEMs – please clarify.

**Editorial Comments:**
- L. 1  Consider "The intrusion of circumpolar deep water in the Amundsen and Bellingshausen Sea Embayments of Antarctica causes ice shelves in the region to melt from below, potentially..." for concision
- L. 7-8 Revise for clarity and concision
- L. 9-11 I recommend reworking these sentences for clarity: "…based on CryoSat-2. Both products show a wide melt channel extending from the grounding line to the ice front, but our high-resolution product indicates..."
- L. 11  "main channel" >> "this channel" since others haven't been introduced yet. Consider "Additionally, it reveals" >> "Additionally, our basal melt rate product reveals…" for specificity
- L. 13  "This emphasizes" >> "These results emphasize" or "Our results emphasize" for specificity
- L. 41  "This approach allowed to assess" >> "This approach allowed the assessment of"
- L. 42  "and thereby also revealing" >> "and thereby revealed" or "…revealing…" for concision
- L. 43  Consider "…satellites to generate high-resolution digital surface models of the PIG ice shelf, which were converted to DEMs by co-registering to…" for concision/readability
- L. 61 "Both" >> "For example,"
- L. 75  "and CryoSat-2 are" >> "and CryoSat-2 surface elevation data are" for specificity
- L. 80-81 "i.e.," >> "e.g.,"

- L. 123 "reference frame" >> "reference surface" to avoid confusion with Lagrangian vs Eulerian or geodetic reference frames
- L. 167-170 Consider "…changes (Sect. 4.3). The latter, along with ice flow divergence (Sect. 4.4), are used in the basal mass balance calculation (Sect. 4.5)" for concision
- L. 246 "calculating the gradients have" >> "calculating velocity gradients have" or as appropriate for specificity
- L. 247 I recommend removing the gradient calculation method types or describing them briefly for clarity
- L. 271 "increasing the spatial resolution" is ambiguous, consider "coarsening" or "refining" as appropriate
- Fig. 7 I recommend including an arrow labeled "direction of flow" on the cross section plots to orient the reader further
- Fig. 8 bottom plots needs x-axis label/units
- 358 "after" >> "downstream of"
- L. 369 Please revise for clarity; see major comment 6
- L. 384 please revise for clarity and concision
- L. 404-405 please revise for clarity

---

## Referee Comment (RC2)

The authors derive maps of elevation change and melt rate from high resolution stereo imagery over a rapidly changing ice shelf in west Antarctica. The paper has two strong axes, one on the methodology addressing challenges pertaining to such datasets and providing a community tool, and the second on the analysis of the findings with regards to ice-ocean interaction making use of an additional modelling dataset.

The paper is well written, clear, and well illustrated. It contains a number of novel elements both methodological and on process that will be of interest to the Cryosphere community. The tool developed by the authors will enable reproduction and should allow further exploitation of the high resolution DEM dataset which should shed light on new processes affecting ice shelves.

I made several comments that I hope can help improve the paper further. My only "major" request to the authors is to strengthen the notion of "resolution", in particular when discussing Lagrangian elevation change and basal melt rate. In several instances a resolution of 50m is mentioned. While this represents the resolution of the original DEMs and the posting of the final product, it might not represent truly the "resolution" of the final product for several reasons. The "resolution" of datasets needed for the mass conservation approaches ranges from 120m for the velocity to several kilometres for the surface dataset. The Lagrangian approach means that effectively over a 7 years period the Lagrangian elevation change and basal melt rate would represent an average over a distance of 2 to 4 km given the speed at which the Dotson ice shelf flows. Finally over distance of 50m or so, hydrostatic equilibrium is unlikely to be realised. While the authors never claim to resolve basal melt features at such a length scale, given the paper's focuses on high resolution inputs and on the production of a 50m "resolution" melt map, they would need to inform the reader and potential users of the product on the maximum effective resolution of the basal melt rate obtained using such an EO based mass conservation approach.

**Other comments:**

L17-22: You may want to add something about the importance of mapping spatially detailed elevation change and melt rate when considering ice shelf and ice sheet stability, e.g. Morlighem et al., 2021; Goldberg et al., 2019

Line 23: replace "or" with ","

Line 24: "remotely through satellite observations of changes in ice shelf surface elevation". Consider rephrasing, melt rate can be calculated that way for ice shelves that display no change in surface elevation. In this mass conservation approach, elevation change (commonly used for Eulerian elevation change) is often a minor term when compared with advection or divergence.

Line 26: "with a temporal resolution defined by field work constrains" note that Apres provides continuous measurements with less ties to "field work constrains"

Lines 38 to 46: Also work by Dutrieux et al, 2013 - https://tc.copernicus.org/articles/7/1543/2013/tc-7-1543-2013.pdf

L46: The term "swath" is not necessarily common knowledge, I would suggest at least adding a reference e.g. Gourmelen et al., 2018.

Lines 48 to 67 on limitations. I would suggest spending a bit more time rephrasing this section. It would be more informative to the readers to have a proper pros and cons of the different approaches with then a focus on what dem-differencing brings to the table. The section on temporal evolution especially needs to be reframed. Altimetry provides ~monthly" systematic observations and has been used to derive time-dependant melt rates e.g. see work by Adusumilli et al., 2020 or Paolo et al., 2022. High res. Stereo-imagery on the other hands are acquired opportunistically with, in general but not always e.g. TDX, a much lower temporal resolution. For mass conservation, the elevation dataset is not the only constrain i.e. ice velocity and information on surface processes are also needed, that will also impact spatial and temporal resolution and accuracy of the final product.

L84: replace "high" with the actual values

L125: Could you specify which geophysical corrections are applied to CryoSat data? It would be useful given the discussion further down about tidal and inverse barometric corrections. Possibly also differentiate those applied to grounded, floating, transition, and importantly what ice shelf mask was used.

L129: Any seasonal variability in the bias?

L136: You would probably need a reference to support the statement of lack of velocity change between 2010 and 2017. Figure 3b of Wild et al., 2023 suggest that areas of slowdown and acceleration exists through Dotson during this period, interestingly matching some of the melt patterns observed including the new marginal channel.

L140: Even in the case of a non-thinning ice shelf SMB would be needed to calculate basal melt from mass conservation, consider rephrasing.

L180: It would be of interest if you were able to comment here on the differences between your inverse barometric correction, and that provided with the CryoSat-2 data L1b product.

L196: "through"?

L245: Indeed this is a critical step in such computation and can result in increased noise in the final product. The authors could add a figure illustrating the improvement of the divergence methods used here.
What is the effective resolution of the final divergence?

L254: Same comment as in line 136. How would the velocity change described in Wild et al. translate into divergence? I am curious also whether the coregistration refinement is robust enough to be used to refine the divergence between DEM dates?

L287: I wonder whether it would not be better to distinguish posting and resolution? Especially when discussing Lagrangian elevation change, and when discussing melt rate.

L307: I can just about see an area of high melt in that sector in Gourmelen et al., 2017, just at the limit of their map. I wonder wether there is a masking issue here rather than a issue with the dataset itself, as the boundary in their map appears pixelated.

Fig. 7: Could you comment on the nature of this melt signal near the calving front, seen in the BURGEE melt map (Figure 5a) but not in the alternative melt dataset shown in figure 5b and 5c.

Fig.8c: x-axis legend is missing

L340 and Fig. 9: Very interesting section. Would it not make more sense, or at least be interesting, to compare and discuss the correlation between the measured melt rate, rather or in addition to the modelled one, with the simulated friction velocity and thermal forcing? Do you seen a similar strong correlation? Where does that correlation breaks down? What does it say about the melt process or about measured melt rate accuracy?

References:

Adusumilli, S., Fricker, H. A., Medley, B., Padman, L., and Siegfried, M. R.: Interannual variations in meltwater input to the Southern Ocean from Antarctic ice shelves, Nature Geoscience, 13, 616–620, https://doi.org/10.1038/s41561-020-0616-z, 2020.

Dutrieux, P., Vaughan, D. G., Corr, H. F. J., Jenkins, A., Holland, P. R., Joughin, I., and Fleming, A. H.: Pine Island glacier ice shelf melt distributed at kilometre scales, The Cryosphere, 7, 1543–1555, https://doi.org/10.5194/tc-7-1543-2013, 2013.

Goldberg, D., Gourmelen, N., Snow, K., Kimura, S., & Millan, R. (2018). How accurately should we model ice shelf melt rates? *Geophysical Research Letters*. https://doi.org/10.1029/2018GL080383

Gourmelen N, Escorihuela M J, Shepherd A, Foresta L, Muir A, Garcia-Mondéjar A, Roca M, Baker S G and Drinkwater M R 2018 CryoSat-2 swath interferometric altimetry for mapping ice elevation and elevation change Adv. Space Res. 62 1226–42

Morlighem, M., Goldberg, D., Dias dos Santos, T., Lee, J., and Sagebaum, M.: Mapping the Sensitivity of the Amundsen Sea Embayment to Changes in External Forcings Using Automatic Differentiation, Geophys. Res. Lett., 48, e2021GL095440, https://doi.org/10.1029/2021GL095440, 2021

Paolo, F., Gardner, A., Greene, C., Nilsson, J., Schodlok, M., Schlegel, N., and Fricker, H.: Widespread slowdown in thinning rates of West Antarctic Ice Shelves, EGUsphere [preprint], https://doi.org/10.5194/egusphere-2022-1128, 2022.

Wild et al., 2023 https://d197for5662m48.cloudfront.net/documents/publicationstatus/120081/preprint_pdf/212e8e698cd6746cfd11628a1e738196.pdf

---

## Author Comment (AC1)

Review for "REMA reveals spatial variability within the Dotson Melt Channel"

The authors use a novel cloud-based method to prepare DEMs and perform basal melt rate calculations called BURGEE (Basal melt rates Using REMA and Google Earth Engine). They show that this method produces high-resolution (50 m) gridded melt rate results that match closely with two other lower-resolution (500 m) studies for large ice shelf features, but that the higher-resolution product captures smaller melt features as well. They also show that errors in basal melt rates due to shifting DEMs using surface velocity maps are reduced at lower resolutions (250 m), without significant loss of spatial variability. Finally, they put forward explanations for some of the spatial variability in surface elevation and melt rates in the vicinity of two basal channels.

This paper fits the topic of the journal and presents a novel, scalable, open-source method for producing basal melt rate maps from REMA DEMs. Overall, I find this study to be well-thought-out, and the figures are *excellent* in both quality and content. However, I have several concerns about the organization and clarity of the paper. I have some recommendations to address these concerns:

Thank you for the review. We have responded to all points clearly marked in red after each comment. The major change that we are suggesting in our response is a reorganization of the methods section, based on your inputs. This includes a brief justification of the different thresholds applied and a more logical order of the different subsections within the methods section including a new figure to help the reader follow the workflow.

1. Several of the thresholds within the steps used to register and correct the REMA strips seem arbitrary to me (Section 4.2) – I would like to see brief justifications for them:
- L. 200 (criterion ii) Why is the 15 m CS-2 vs REMA difference used to filter CS-2 outliers? I see how this filter would remove control points where either CS-2 or REMA is erroneous, but if the strip is consistently 15+ m too high/low but otherwise fine, it will be excluded. If the authors wish to remove erroneous CS-2 points before using them for registration, I recommend filtering CS-2 using published quality flags and/or by comparing CS-2 to the published REMA mosaic and/or ICESat-2 and removing outliers that fall outside some range around a spatially-averaged elevation or elevation difference (for example, interpolate REMA mosaic to CS-2 points, take moving mean of both sets of points or their differences over some along-track distance, and remove CS-2 elevations that are above/below 2 standard deviations from the moving mean) – this would likely be reported in Section 3.1.
- Line 202 (criterion iii) Why is the minimum threshold for the number of match-ups 80? It seems that this minimum should depend on the relative size and resolution of the strip and footprint of CS-2.
- L 203-205 (criterion iv) Similar to my comment above, it is not clear why the north-south and east-west distances between distal CS-2 points should be >40 and 10 km, respectively. This is effectively setting a minimum size for the strips – and the strips could easily be filtered earlier by area or length and width. I think this step is fine, but please include a specific justification for the minimum distances. It would also be helpful to include a figure showing the CS-2 reference ground tracks, or actual ground tracks used in this study, and how they are oriented compared to the strips (for example, add the reference ground tracks to Fig. 3h), and a discussion of how these orientations informed the thresholds that have been set.

In the updated manuscript we will provide a brief justification of the different thresholds which we also outline here. Both the REMA strips and the CryoSat-2 elevations are filtered based on the REMA mosaic. This was not clear in the original manuscript but will be made clear in the updated version. Furthermore, CS-2 points are also filtered using the quality flags provided by ESA which we will make sure to mention in the updated manuscript. Regarding the 15 m CS-2 vs REMA strip difference criteria we have investigated this a bit further and found that the criteria has almost no effect, since it rarely happens that CS-2 points fall out of this criteria. We have therefore decided to leave this criteria out. Regarding the minimum threshold for the number of match-ups it is a balance between good quality co-registration and keeping a sufficient number of REMA strips. Lastly, regarding the north-south and east-west distance threshold, the actual threshold applied is 60 km north-south and 10 km east-west, which will be updated accordingly in the updated manuscript. Applying a latitude-longitude threshold is essentially the same as applying a length-width threshold due to the nature of the REMA strips. We are aware that our conservative threshold may filter out good quality REMA strips that are smaller than 60x10 km, but in this way we ensure the best possible quality, which is crucial when the goal is to resolve small-scale features.
Finally, regarding a figure showing the CS-2 reference ground tracks we do not believe that this is relevant. Below is a figure of the CS-2 ground tracks over Dotson. This however, does not correspond to the position of the actual CS-2 measurements due to the point-of-closest-approach which radar altimetry is subject to. So the orientation of the CS-2 tracks have not been used to inform the thresholds and they also cannot be used for that since they do not represent the location of the actual CS-2 data.

[Figure]

Developed by the Norwegian Polar Institute. Visit npolar.no/quantarctica

2. The process by which the annual DEM mosaics shown in Fig. 3 are produced is not outlined clearly. I recommend including a separate section in the methods addressing this, separately from the (co-)registration of each strip. This omission also leads to confusion in interpreting the basal melt rate results in Fig. 5, which show maps with nearly complete coverage for combinations of years in which the mosaics are not complete. How are the complete coverage maps in Fig. 5 obtained? Are supplementary strips used that were not included in the mosaics?

We can see why confusion may occur around section 4.2-4.6. We will therefore rearrange these sections to the following: 4.2 Co-registration (from which the "final result" are the co-registered DEM strips), 4.3 Eulerian elevation change (for which the DEM strips are used), 4.4 Divergence of the velocity field, and 4.5 Lagrangian elevation change and basal mass balance (this is where the strips are turned into yearly DEM mosaics for computational reasons). In section 4.2 we will still include a figure similar to Fig. 3 to show the yearly DEM coverage, since the yearly DEM strips coverage is the same as the yearly DEM mosaic coverage. In section 4.5 we will include the below figure to illustrate how the Lagrangian displacement is performed and thereby also how the yearly DEM mosaics are created. This figure will also help illustrate how the basal mass balance is calculated, both when talking about a 3-yearly period and the full trend. On top of that we will also modify the text accordingly, and make the process clearer by referring to different steps in this new figure. These changes to the manuscript should not only answer this point, but also point 3 and 4 below.

[Figure]

**A.** Co-registered DEM strips from one year

**B.** Add bands to strips: Divergence Nearest 10-daily firn air content

**C.** Displace using ITS_LIVE to 1 January of year $t_{end}$

**D.** Add a time stamp band from every strip

**E.** Create DEM mosaic using the *quality mosaic function*

**F.** Add ice flow divergence band: $(h - h_f)(\nabla \cdot \boldsymbol{u})$

**G.** Feature tracking displacement with respect to the DEM mosaic of $t_{end}$

**H.** Perform steps A-G on all years considered (either 3-yearly or full trend)

**I.** Use elevation and time stamp bands to calculate the Lagrangian elevation change of the period

**J.** Use firn air content and time stamp bands to calculate the temporal change of firn air content of the period

**K.** Use the time stamp bands to access the first and last date and apply that period to calculate the average yearly SMB from the monthly fields

**L.** Take the mean of all ice flow divergence bands to get the mean ice flow divergence

**M.** Use the Lagrangian elevation change, temporal change of firn air content, average yearly SMB, and mean ice flow divergence to calculate the basal melt rate over the period using eq. (4)

3. Related to point 2, the description in Section 4.3 of the Lagrangian displacement method is unclear. I made connections (perhaps mistakenly) between both a. displacing strips acquired within a given austral summer as part of the co-registration process to produce an annual mosaic, and b. displacing annual mosaics to t_end in order to calculate rates of elevation change. However, the distinction between these processes is not clear as written, and I think that devoting a section to describing the mosaicking process will greatly help readability. It may also help to add a panel to Fig. 2 showing the workflow for the mosaicking.

See our response to point 2 above.

4. I recommend reorganizing section 4.5 for clarity, particularly to distinguish the different time scales used in basal mass balance estimates. Please see major comments 2 and 3 above about how DEM mosaics are created and how DEM strips/mosaics are displaced to estimate Lagrangian changes. I find a number of possible inconsistencies that could be clarified by separating bits from Sections 4.2 and 4.3 to form a new section devoted to describing how the annual mosaics are made:

If we consider an example in which we calculate a tri-yearly basal melt rate between the years 2011/12 – 2014/15:

- "$M_s$ is the average yearly surface mass balance from the time of the first DEM mosaic to the time of the last DEM mosaic" indicates that the surface mass balance is the average value of all annual values between 2011/2012 and 2014/15.
o  However, section 3.3 indicates that monthly SMB values are used – so is the 2011/2012 annual DEM associated with the average of all monthly values between 1 July 2011 – 30 June 2012 OR is each strip associated with $M_s$ most close to its collection time, and the $M_s$ is "mosaicked" similarly as the strips to create an annual $M_s$ map?? Please clarify here and/or in Section 3.3/new mosaicking section.

- "The divergence field and the firn air content are added as bands to the DEMs before applying the velocity displacement" indicates that The 2011/12 annual mosaic and its firn air content and divergence field are shifted to the ice's position in 2014/15.
o  However, Section 3.4 indicates that firn air content is obtained on a 10-daily basis – so is the 2011/2012 annual DEM associated with the average of all 10 day values between 1 July 2011 – 30 June 2012? OR is each strip associated with the firn air content most close to its collection time, and the firn air content is "mosaicked" similarly as the strips to create an annual firn air content map? Please clarify here and/or in Section 3.4/new mosaicking section.

- L. 261 indicates that (h − hf )($\nabla$ · u) is calculated on a yearly basis – so is each annual DEM associated with its own (h − hf )($\nabla$ · u) map? Please clarify here and/or in Section 4.4/new mosaicking section.
o  Related, "If $\Delta t$ > 1 year, the average of the years considered is taken" – does this mean that for 2011/12 – 2014/15

that the (h − hf )($\nabla \cdot$ u) used in Eq. 1 is the mean of the 2011/12 (h − hf )($\nabla \cdot$ u) map and the 2014/15 (h − hf )($\nabla \cdot$ u) map? Please clarify.

- "The firn air content and divergence fields are therefore also displaced, and their values correspond to those at the original position and time (in the case of firn air content) of the DEM strips" – this indicates that divergence fields and firn air contents are calculated/interpolated to each strip before annual DEM mosaic are created. I recommend this be moved to a new section about how the annual DEM mosaics are created.

- L. 266-269 about the entire study period relates to later questions about how the maps in Fig. 4 are produced. I read that all annual mosaics are shifted to their position in 2017/18, moving ice parcels forward one year at a time, and recalculating (h − hf )($\nabla \cdot$ u) after each 1-year shift – is this right? Please clarify if not. Please also clarify how Mb is calculated over the entire time period – are elevation changes, firn air column averages, and $M_s$ averages computed between every combination of flow-shifted DEMs (i.e. 2010/11 – 2011/12 and 2010/11 – 2012/13… and …2010/11 – 2017/18, and so on), or between consecutive DEMs (i.e. 2010/11 – 2011/2012 and 2011/12 – 2012/13 and so on), OR between each DEM and the 2017/18 DEM (i.e. 2010/11 – 2017/18 and 2011/12 – 2017/18 and so on)? Is the map reported in Fig. 4 the average or median across all combinations, or something else?

Please refer to our response to point 2.

5. Currently, some contextual discussion is present in the Results section, (e.g. how main channel's bend may be caused by buoyancy forcing of a meltwater plume; how variable melting pinning points may cause surface undulations; postulation about how the hydrostatic assumption might obscure the western margin melt channel in other studies), which makes parts of the Discussion seem like a recap of the results. I recommend reorganizing the Results and Discussion sections a bit so that the Results presents only the authors' novel findings, while the discussion places them in context.

As suggested we will move the contextual discussion which is present in the Results (L313 - L314 and L331 - L334) to the Discussion.

6. Please check for consistency when describing the locations of ice shelf features. I recommend explicitly naming the two basal channels discussed to remove ambiguity (for example, abbreviate the large, more central basal channel DMC (Dotson Main Channel) and the smaller channel at the western shear margin DWC (Dotson Western Channel) or similar). I recommend including a brief section on the morphology of the entire ice shelf, either in the Introduction or beginning of the Results (since a major result of this study is high-resolution elevation maps) in which the features can be identified. They could also be marked on this paper's Fig. 1 with superimposed lines of different colors/weights/styles as in Fig. 1 of Dow et al. (2018, Science Advances – Nansen ice shelf channels). This will make it easier for the reader to orient themselves throughout the paper.

We will name the channels Dotson Main Channel and Dotson Coastal Channel and be consistent with that throughout the paper. We do not agree that high resolution elevation maps are a major result of this study, but rather what we use them for, namely the high resolution basal melt rates. Further, surface depressions in an ice shelf does not necessarily mean high basal melt rates which we also see on Dotson. In the Figure below we have marked an anchor shaped surface feature on the ice shelf which is not associated with high melt rates. These kinds of features are very interesting, especially because we do not know what has formed this feature, but it is out of the scope of this paper. Therefore, we also believe that adding a section on the morphology of the ice shelf will reduce the readability of the paper, since our focus is not on surface features, but on melt features from which some happen to be explained by surface features.

[Figure]

**Minor scientific comments:**

- L. 22 I recommend being more specific about what the potential consequences of basal melting and ice shelf thinning may be (e.g. ice shelf/ice sheet mass loss and Sea level rise)

We will add the following line in L21: "...be initiated (Schoof, 2007; Ritz et al., 2015). Furthermore, Morlighem et al., 2021 show that the location of temporal changes in basal melt of ice shelves in the Amundsen sea sector of Antarctica matters for glacier-wide mass balances, making spatially detailed elevation changes and basal melt rates important. Therefore, …"

- L. 49-50 Most literature suggests that this loss-of-lock near steep gradients is unique to radar altimetry – I recommend including a citation here

We will include the following citation: Dehecq A, Gourmelen N, Shepherd A, Cullen R and Trouvé E 2013 Evaluation of CryoSat-2 for height retrieval over the Himalayan range CryoSat-2 third user workshop (Dresden, Germany)

- L. 50 At this point it is not clear why Fig. 1 is included, since there is no indication so far in the introduction that CS-2 data were used in this study, or that this study focuses on Dotson. I recommend moving L. 50 from "From Fig. 1…" to 54 to the paragraph starting in L. 84, including a brief reminder of the limitations of radar altimetry on steep slopes

We will move the suggested part to the paragraph starting in L84 and adjust it accordingly.

- L. 55-57 The impact of the resolution is relative to what questions are being answered – I recommend being more specific about what is lost due to varying resolutions across studies/methods. Furthermore, in L. 56 "the chosen temporal and spatial resolution" is a little misleading, because the spatiotemporal resolution and range is constrained by data availability, not just choice – please revise.

L48-67 will be re-written to have clear proper pros and cons of the different methods. We will pay extra attention to the discussion of temporal and spatial resolution as suggested here and by the other reviewer.

- L. 78/L. 354 Chartrand & Howat used OIB for registration where available and CS-2 otherwise (ICESat and REMA do not overlap temporally) – please revise

L78 will be changed to "Chartrand and Howat (2020) used Operation IceBridge where available to co-register the REMA strips, and CryoSat-2 otherwise."
L353/354 will be changed to "The use of CryoSat-2 and Google Earth Engine in BURGEE allows to process significantly more REMA strips than if using Operation IceBridge alone, and it allows for fast computations."

- L. 83 Please specify what metric (elevation, elevation change, basal melt rate) the signal-to-noise ratio is concerning for

"...and signal-to-noise ratio of both the Lagrangian elevation change and basal melt rate." will be added to L83.

- L. 103 Please specify the signs for melt/accumulation in Eq. 1

"...and M_b is the basal mass balance, defined as positive for melt." will be added to to L103

- L. 122-131 I recommend including some or all of the detail about how both CS-2 and REMA strips are filtered prior to registration (including moving L. 173-174 to here).

In L127 we will add the following: "Using the provided data from CryoSat-2, elevations are corrected for the following geophysical parameters regardless of the surface: ocean loading tide, solid earth tide, geocentric polar tide, dry and wet tropospheric correction, and ionospheric correction. Ice shelf specific corrections are outlined in Sect. 4.1."
Furthermore, the filtering based on the REMA mosaic will be moved to this section including the correction for the geoid as proposed.

- L. 154/Section 3.5 Consider renaming this "basal melt rate comparison products" or similar since "evaluation" connotes methodology.

We will rename this section to "Basal melt rate comparison products" as suggested.

- L. 173-174 I recommend moving the sentence beginning "However, before all that is done…" this to Sect. 3.1 and revising surrounding sentences as appropriate

L173-175 about filtering based on the REMA mosaic and correcting for the geoid will be moved to Sect. 3.1.

- L. 179-182 Applying the tide and ibe corrections requires knowledge of the time the elevation data were collected – please specify how the collection times of the strips were determined and discuss the implications of using 6-hourly corrections

In L179 we will change to: "Tidal heights are obtained from the CATS2008 model on a 6-hourly interval at a ~3 km spatial resolution for the REMA strips. Tidal heights for CryoSat-2, however, are obtained at their point locations and acquisition times. Possible implications of using 6-hourly tides for the REMA strips are thereby taken care of when co-registering to CryoSat-2."
In L182 we will add the following: "...effect (Wunsch, 1972). The acquisition time used for the REMA strips to correct for tide and inverse barometer effect is that of sourceImage1 from the metadata, meaning the acquisition time of the first stereo image."

- L. 186 Please specify which grounding line product was used to calculate the correction transition

In L191 we will add the following: "For this purpose we use the ASAID grounding line (Fig. 1, Bentley et al., 2014)."

- L. 190/Section 4.2 Please see major comments 1 & 2 above
Please refer to our responses to point 1 and 2 above.

- L. 214 Please provide brief justification for using tri-yearly basis for elevation changes – why not annual?

We will add: "...throughout the study period. A tri-yearly period is chosen due to the limited to no DEM coverage in 2011/12 and 2012/13 (see Fig. 3), alongside with the limited coverage of the center part of the ice shelf in 2015/16 and 2017/18 (see Fig. 3).

- L. 217-218, L. 244 Similar to major comment 1 above, please provide brief justification for removing points with elevation change rate of > 15 m/yr as opposed to deviation from a spatial mean. Particularly in Lagrangian, this could obscure fast, small scale processes like rift opening (which may be fine for this study since it is focused on melt rates and not fracture)

In L218 we will add: "This filter could possibly filter out fast, small scale processes like rift opening, which in this study is acceptable due to our focus on the basal melt rates."

- L. 224-241 Please see major comment 3 above

Please see the response to comment 2 above.

- L. 256-270/Section 4.5 Please see major comment 4 above

Please see the response to comment 2 above.

- L. 275 Related to major comments 2-4 above, please define what a "perfectly aligned annual DEM mosaic" is. I would recommend avoiding using "perfectly" unless it is specifically defined earlier.

We will remove the word "perfectly".

- L. 283 Please include more detail about the various resolutions used and why

We will adjust to: "...changes is then calculated at a 50 m, 100 m, 250 m, and 500 m posting to see what posting is required for the artificial error to cancel out. The 500 m posting corresponds to what has been used when using CryoSat-2 alone (Gourmelen et al., 2017) and therefore serves as our most coarse limit given that we apply elevation data of much higher resolution."

Note here that we in the updated manuscript will distinguish between posting and resolution.

- L. 286/Fig. 4 Related to major comment 4 above, please describe how the 2010/11 – 2017/18 trends were calculated

Please refer to our response to point 2 above. The updated section 4.5 including the new figure should describe this process clearly.

- L. 293 Related to comment on L. 186 above – please describe how the Dotson/Crosson border was defined and/or why was the ASAID grounding line is used to define the Dotson ice shelf

We use the Dotson Ice Shelf as a test site to show what BURGEE is capable of, for which the ASAID grounding line is used to define the Dotson/Crosson border. We acknowledge that newer and better grounding line products exist for the Dotson/Crosson ice shelf/shelves, but they do not change the findings of this paper.

- L. 294-295 isn't the "smaller area" the same as "at the border towards the Crosson" in L. 293? Remove or revise.

Correct. Thanks. We will remove this last sentence.

- L. 299 Specify which panels of Fig. 5 the striped pattern is visible

We will add: "...on some of the 3-yearly maps (e.g. 2011/12-2014/15, Fig. 5e and 2012/13-2015/16, Fig. 5f) due..."

- L. 300 varying coverage – see major comment 2

See our reply above to comment 2.
Here in L300 we will also adjust to: "The varying coverage also implies that the trend is taken over different time periods whenever there is missing data in the DEM mosaics, this means that we calculate the basal melt rate in a pixel if there is data from at least two of the yearly DEM mosaics."

- L. 303 spatial description – see major comment 6

Will be adjusted to: "...present along the Dotson Main Channel (red arrow),…"

- L. 313 "bending toward the margin" – which margin? See major comment 6

Will be changed to: "bending towards the western margin"

- L. 315 I recommend including a velocity and/or strain rate map in one of the figures so the reader can see the convergence zone clearly

The convergence zone is a melt convergence zone, not an ice flow convergence zone. We will clarify this in the manuscript.

- L. 321-322 see major comment 6

See our response to comment 6.

- L. 324 Please specify what the cross section is and what quantity (basal melt, thinning?) has an error of +- 2 m/yr

"resulting error" will be replaced with "basal melt rate difference"

- L. 330 please specify what quantity (elevation, melt rate?) has a smooth/wavy pattern

"the main channel" will be changed to "the basal melt within the Dotson Main Channel"

- L. 338-339 Please clarify what is meant by "implying greater thicknesses", or consider reframing this as evidence for deeper basal drafts experiencing more melt, which is seen elsewhere in literature

A greater ice thickness and deeper basal draft is the same, but we will change it to "implying deeper basal drafts, and vice versa."

L. 362-363 Alley et al. (2019, Sci. Adv.) also shows marginal channels observationally, and Alley et al., (2016, Nat. Geosc.) identify this western marginal channel (not plotted, but in the shapefiles)

We are not sure exactly what channel is referred to in the comment here since L362-363 discusses the Dotson Main Channel, but it sounds like that the reviewer is referring to the Dotson Coastal Channel. We do not have access to the shapefiles at the moment, but will ask for access to them.

L. 381 It seems that the authors should be able to verify the presence of a surface depression associated with the basal channel from the DEMs – please clarify.

We will add the following sentence after the existing one in L381: "The presence of the channel is further supported by a surface depression in the DEMs (see Fig. 3a,d-f)."

**Editorial Comments:**
- L. 1 Consider "The intrusion of circumpolar deep water in the Amundsen and Bellingshausen Sea Embayments of Antarctica causes ice shelves in the region to melt from below, potentially..." for concision

Will do.

- L. 7-8 Revise for clarity and concision

Will be changed to: "We present a novel method: Basal melt rates Using REMA and Google Earth Engine (BURGEE). The high resolution of BURGEE is supported through a sensitivity study of the Lagrangian displacement."

- L. 9-11 I recommend reworking these sentences for clarity: "…based on CryoSat-2. Both products show a wide melt channel extending from the grounding line to the ice front, but our high-resolution product indicates..."

Will do.

- L. 11 "main channel" >> "this channel" since others haven't been introduced yet. Consider "Additionally, it reveals" >> "Additionally, our basal melt rate product reveals…" for specificity

What we will do: "main channel" >> "this channel" and "Additionally, it reveals" >> "Additionally, BURGEE reveals…"

- L. 13 "This emphasizes" >> "These results emphasize" or "Our results emphasize" for specificity

Will change to "These results emphasize"

- L. 41 "This approach allowed to assess" >> "This approach allowed the assessment of"

Will do.

- L. 42 "and thereby also revealing" >> "and thereby revealed" or "…revealing…" for concision

Will change to "and thereby revealed"

- L. 43 Consider "…satellites to generate high-resolution digital surface models of the PIG ice shelf, which were converted to DEMs by co-registering to…" for concision/readability

Will do.

- L. 61 "Both" >> "For example,"

Will do.

- L. 75 "and CryoSat-2 are" >> "and CryoSat-2 surface elevation data are" for specificity

Will do.

- L. 80-81 "i.e.," >> "e.g.,"

Will do.

- L. 123 "reference frame" >> "reference surface" to avoid confusion with Lagrangian vs Eulerian or geodetic reference frames

Will do. As mentioned previously we will also move L173-175 to here.

- L. 167-170 Consider "…changes (Sect. 4.3). The latter, along with ice flow divergence (Sect. 4.4), are used in the basal mass balance calculation (Sect. 4.5)" for concision

Will do.

- L. 246 "calculating the gradients have" >> "calculating velocity gradients have" or as appropriate for specificity

Will do.

- L. 247 I recommend removing the gradient calculation method types or describing them briefly for clarity

Please refer to the new structure of the methods section, which also includes a move of this section.

- L. 271 "increasing the spatial resolution" is ambiguous, consider "coarsening" or "refining" as appropriate

Will change to "refining".

- Fig. 7 I recommend including an arrow labeled "direction of flow" on the cross section plots to orient the reader further

Thanks, we will do that.

- Fig. 8 bottom plots needs x-axis label/units

Thanks. Will be fixed.

- 358 "after" >> "downstream of"

Will change.

- L. 369 Please revise for clarity; see major comment 6

We will do this: "eastward convergence zone" >> "eastward melt convergence zone"

- L. 384 please revise for clarity and concision

"Furthermore, we can assess temporal changes on a 3-yearly basis, though it should be noted that a shorter time frame puts constrains on the spatial coverage due to the yearly REMA coverage." >> "Furthermore, we can assess temporal changes on a 3-yearly basis, but with poorer coverage than the full trend due to the yearly REMA coverage."

- L. 404-405 please revise for clarity

"We perform a sensitivity study which further supports the trustworthiness of the observed small-scale features and indicates that a 50 m spatial resolution is feasible." >> "We perform a sensitivity study which supports the trustworthiness of the observed small-scale features. It further indicates that a 50 m spatial posting is feasible. "

---

## Author Comment (AC2)

The authors derive maps of elevation change and melt rate from high resolution stereo imagery over a rapidly changing ice shelf in west Antarctica. The paper has two strong axes, one on the methodology addressing challenges pertaining to such datasets and providing a community tool, and the second on the analysis of the findings with regards to ice-ocean interaction making use of an additional modelling dataset.
The paper is well written, clear, and well illustrated. It contains a number of novel elements both methodological and on process that will be of interest to the Cryosphere community. The tool developed by the authors will enable reproduction and should allow further exploitation of the high resolution DEM dataset which should shed light on new processes affecting ice shelves.

I made several comments that I hope can help improve the paper further. My only "major" request to the authors is to strengthen the notion of "resolution", in particular when discussing Lagrangian elevation change and basal melt rate. In several instances a resolution of 50m is mentioned. While this represents the resolution of the original DEMs and the posting of the final product, it might not represent truly the "resolution" of the final product for several reasons. The "resolution" of datasets needed for the mass conservation approaches ranges from 120m for the velocity to several kilometres for the surface dataset. The Lagrangian approach means that effectively over a 7 years period the Lagrangian elevation change and basal melt rate would represent an average over a distance of 2 to 4 km given the speed at which the Dotson ice shelf flows. Finally over distance of 50m or so, hydrostatic equilibrium is unlikely to be realised. While the authors never claim to resolve basal melt features at such a length scale, given the paper's focuses on high resolution inputs and on the production of a 50m "resolution" melt map, they would need to inform the reader and potential users of the product on the maximum effective resolution of the basal melt rate obtained using such an EO based mass conservation approach.

Thank you for the review. We have responded to all points clearly marked in red after each comment.
In the updated manuscript we will make sure to distinguish between posting and resolution and thereby write that we offer our product at a 50 m posting, and of course be consistent with this throughout the paper. We will make the difference between resolution and posting clear and clearly state that this also implies that melt signals in our product at very small length scales are not necessarily true signal.

Other comments:
L17-22: You may want to add something about the importance of mapping spatially detailed elevation change and melt rate when considering ice shelf and ice sheet stability, e.g. Morlighem et al., 2021; Goldberg et al., 2019

We will add the following line in L21: "...be initiated (Schoof, 2007; Ritz et al., 2015). Furthermore, Morlighem et al., 2021 show that the location of temporal changes in basal melt of ice shelves in the Amundsen sea sector of Antarctica matters for glacier-wide mass balances, making spatially detailed elevation changes and basal melt rates important. Therefore, …"

Line 23: replace "or" with ","

Will do

Line 24: "remotely through satellite observations of changes in ice shelf surface elevation". Consider rephrasing, melt rate can be calculated that way for ice shelves that display no change in surface elevation. In this mass conservation approach, elevation change (commonly used for Eulerian elevation change) is often a minor term when compared with advection or divergence.

Will be rephrased to: "remotely through satellite observations of changes in ice shelf surface elevation in combination with information about ice flow and surface processes."

Line 26: "with a temporal resolution defined by field work constrains" note that Apres provides continuous measurements with less ties to "field work constrains"

Will be rephrased to: "with a temporal resolution defined by field work constrains, though it should be noted that autonomous phase sensitive radars provide continuous point measurements with less ties to field work constrains."

Lines 38 to 46: Also work by Dutrieux et al, 2013 -
https://tc.copernicus.org/articles/7/1543/2013/tc-7-1543-2013.pdf

We will add the following in L46: "Earlier, also Dutrieux et al 2013 have assessed the basal melt rate of the Pine Island Glacier Ice Shelf using a similar approach, but using the slightly coarser resolution SPIRIT DEMs."

L46: The term "swath" is not necessarily common knowledge, I would suggest at least adding a reference e.g. Gourmelen et al., 2018.

We will add Gourmelen et al., 2018 as a reference.

Lines 48 to 67 on limitations. I would suggest spending a bit more time rephrasing this section. It would be more informative to the readers to have a proper pros and cons of the different approaches with then a focus on what dem-differencing brings to the table. The section on temporal evolution especially needs to be reframed. Altimetry provides ~monthly" systematic observations and has been used to derive time-dependant melt rates e.g. see work by Adusumilli et al., 2020 or Paolo et al., 2022. High res. Stereo-imagery on the other hands are acquired opportunistically with, in general but not always e.g. TDX, a much lower temporal resolution. For mass conservation, the elevation dataset is not the only constrain i.e. ice velocity and information on surface processes are also needed, that will also impact spatial and temporal resolution and accuracy of the final product.

In the updated manuscript we will rephrase lines 48-67. Line 51-54 will be moved to the paragraph starting at L84. We will further do as proposed and have a proper pros and cons of the different approaches.

L84: replace "high" with the actual values

Will be rephrased to: "... to obtain high spatial (50 m) and temporal (3-yearly) resolution thinning and basal…"

L125: Could you specify which geophysical corrections are applied to CryoSat data? It would be useful given the discussion further down about tidal and inverse barometric corrections. Possibly also differentiate those applied to grounded, floating, transition, and importantly what ice shelf mask was used.

In L127 we will add the following: "Using the provided data from CryoSat-2, elevations are corrected for the following geophysical parameters regardless of the surface: ocean loading tide, solid earth tide, geocentric polar tide, dry and wet tropospheric correction, and ionospheric correction. Ice shelf specific corrections are outlined in Sect. 4.1."
In L191 we will add the following: "For this purpose we use the ASAID grounding line (Fig. 1, Bentley et al., 2014)."

L129: Any seasonal variability in the bias?

We did not look into this since the REMA strips availability more or less is constrained to the austral summer period.

L136: You would probably need a reference to support the statement of lack of velocity change between 2010 and 2017. Figure 3b of Wild et al., 2023 suggest that areas of slowdown and acceleration exists through Dotson during this period, interestingly matching some of the melt patterns observed including the new marginal channel.

It is not entirely clear from the Wild et al., 2023 manuscript how the velocities are obtained. The ITS_LIVE velocities over Dotson are incomplete up until 2013. However, when we calculate the trend of all yearly ITS_LIVE

velocities from 2013-2018 and applying roughly the same colour scheme as Wild et al Fig. 3 we see the following pattern:

[Figure]

Here we see an almost uniform pattern of a very slight acceleration. Further, it seems that the velocity changes mentioned in Wild et al happened before our study period began. In the updated manuscript we will add a reference to the Lilien et al., 2018 paper who argues that Dotson was stable: "Dotson, which has maintained its speed despite increasingly high melt rates near its grounding line"

L140: Even in the case of a non-thinning ice shelf SMB would be needed to calculate basal melt from mass conservation, consider rephrasing.

Rephrased to: Since part of the ice shelf mass balance may be related to surface processes ( Ms in eq. (4)) monthly surface mass balance values are obtained from the regional climate model RACMO 2.3p3 (van Wessem et al., 2018).

L180: It would be of interest if you were able to comment here on the differences between your inverse barometric correction, and that provided with the CryoSat-2 data L1b product.

The inverse barometer correction is only provided to the CryoSat-2 SAR data (open ocean) and not the SARin mode which is used around the coastline of Antarctica.

L196: "through"?

Yes! - Changed

L245: Indeed this is a critical step in such computation and can result in increased noise in the final product. The authors could add a figure illustrating the improvement of the divergence methods used here.
What is the effective resolution of the final divergence?

The improvement resulting from this method has already been nicely illustrated by Berger et al., 2017 Fig. 3. Due to the nature of the regularized divergence, namely the regularization parameter \alpha in Chartrand 2011, it is not possible to determine the effective resolution of the final divergence.

L254: Same comment as in line 136. How would the velocity change described in Wild et al. translate into divergence? I am curious also whether the coregistration refinement is robust enough to be used to refine the divergence between DEM dates?

As seen in the figure above we do not see the same trend/velocity change as Wild et al. when using all yearly ITS_LIVE velocities from 2013-2018. The trend pattern which we can see in the above pattern has a speckled pattern due to noise in the yearly velocities, noise which is not present to the same extent in the averaged

ITS_LIVE product. There are no sharp edges between areas of deceleration and acceleration, which is why the resulting signal in the divergence field due to changing velocities will be limited.

Whether or not the coregistration refinement is robust enough to refine the velocities and thereby also the divergences as well is a good question. However, in our study, the final correction / feature tracking is never larger than 300 m, and often also much less than that. So given the 120 m surface velocity resolution, the extra correction which the feature tracking may provide will most likely be too small to properly affect the divergence.

L287: I wonder whether it would not be better to distinguish posting and resolution?
Especially when discussing Lagrangian elevation change, and when discussing melt rate.

In the updated manuscript we will make sure to distinguish between posting and resolution, and therefore say that we provide our results at 50 m posting instead of resolution. We will make sure to be consistent with this throughout the paper.

L307: I can just about see an area of high melt in that sector in Gourmelen et al., 2017, just at the limit of their map. I wonder wether there is a masking issue here rather than a issue with the dataset itself, as the boundary in their map appears pixelated.

In Figure 5 in Lambert et al. 2022 the light gray area shows the "missing pixels" of Gourmelen et al., 2017. Here it can be seen that it is a narrow band of pixels which is missing compared to LADDIE, and that most of the area where the channel is present is indeed an area captured by Gourmelen et al., 2017. Therefore, we think that it may be a consequence followed by CryoSat-2 limitations in this area of the ice shelf.

Fig. 7: Could you comment on the nature of this melt signal near the calving front, seen in the BURGEE melt map (Figure 5a) but not in the alternative melt dataset shown in figure 5b and 5c.

In L306 we will add the following: "A slight exception to this is the melt signal near the calving front seen in BURGEE. Here, there are large crevasses and fractures in the ice shelf, which may not be well represented in the divergence signal when assessing the basal melt rate at a 50 m posting. "

Fig.8c: x-axis legend is missing

Thanks! Has been fixed.

L340 and Fig. 9: Very interesting section. Would it not make more sense, or at least be interesting, to compare and discuss the correlation between the measured melt rate, rather or in addition to the modelled one, with the simulated friction velocity and thermal forcing? Do you seen a similar strong correlation? Where does that correlation breaks down? What does it say about the melt process or about measured melt rate accuracy?

That would indeed be interesting, but since LADDIE is forced with a different ice shelf geometry (BedMachine) a 1-1 comparison between the modeled friction velocity and thermal forcing and the BURGEE basal melt rates cannot be done. In the updated manuscript we will make sure to mention this.

References:

Adusumilli, S., Fricker, H. A., Medley, B., Padman, L., and Siegfried, M. R.: Interannual variations in meltwater input to the Southern Ocean from Antarctic ice shelves, Nature Geoscience, 13, 616–620, https://doi.org/10.1038/s41561-020-0616-z, 2020.

Dutrieux, P., Vaughan, D. G., Corr, H. F. J., Jenkins, A., Holland, P. R., Joughin, I., and Fleming, A. H.: Pine Island glacier ice shelf melt distributed at kilometre scales, The Cryosphere, 7, 1543–1555, https://doi.org/10.5194/tc-7-1543-2013, 2013.

Goldberg, D., Gourmelen, N., Snow, K., Kimura, S., & Millan, R. (2018). How accurately should we model ice shelf melt rates? Geophysical Research Letters. https://doi.org/10.1029/2018GL080383

Gourmelen N, Escorihuela M J, Shepherd A, Foresta L, Muir A, Garcia-Mondéjar A, Roca M, Baker S G and Drinkwater M R 2018 CryoSat-2 swath interferometric altimetry for mapping ice elevation and elevation change Adv. Space Res. 62 1226–42

Morlighem, M., Goldberg, D., Dias dos Santos, T., Lee, J., and Sagebaum, M.: Mapping the Sensitivity of the Amundsen Sea Embayment to Changes in External Forcings Using Automatic Differentiation, Geophys. Res. Lett., 48, e2021GL095440, https://doi.org/10.1029/2021GL095440, 2021

Paolo, F., Gardner, A., Greene, C., Nilsson, J., Schodlok, M., Schlegel, N., and Fricker, H.: Widespread slowdown in thinning rates of West Antarctic Ice Shelves, EGUsphere [preprint], https://doi.org/10.5194/egusphere-2022-1128, 2022.

Wild et al., 2023 https://d197for5662m48.cloudfront.net/documents/publicationstatus/120081/preprint_pd f/212e8e698cd6746cfd11628a1e738196.pdf

Lambert, E., Jüling, A., Wal, R. S. W. V. D., and Holland, P. R.: Modeling Antarctic ice shelf basal melt patterns using the one-Layer Antarctic model for Dynamical Downscaling of Ice – ocean Exchanges ( LADDIE ), The Cryosphere Discuss. [preprint], 2022, 1–39, https://doi.org/10.5194/tc-2022-225, 2022.

Chartrand, R.: Numerical Differentiation of Noisy, Nonsmooth Data, ISRN Applied Mathematics, 2011, 1–11, 445 https://doi.org/10.5402/2011/164564, 2011.

Lilien, D. A., Joughin, I., Smith, B., and Shean, D. E.: Changes in flow of Crosson and Dotson ice shelves, West Antarctica, in response to elevated melt, The Cryosphere, 12, 1415–1431, https://doi.org/10.5194/tc-12-1415-2018, 2018.

---

## Author Response (AR1)

Dear Christian Haas,

Thank you very much for the feedback.
Comments from the reviewers that have not been included explicitly are outlined below. In black is the comment from the reviewer, in red our original answer and in blue our updated answer/comments.
Furthermore, we have in the meantime found a bug in the code which has now been fixed. The ice flow divergences had not been adjusted accordingly for their original resolution. The consequence of this was that areas with high divergence or convergence were suffering from either too high or too low basal melt rates. This means that the coastal channel discussed in the original manuscript (Fig. 6 in original manuscript) is less pronounced in the updated results. We have therefore decided to exclude this part from the updated manuscript.

Kind regards,
Ann-Sofie P. Zinck and co

Reviewer 1 comments:

6. Please check for consistency when describing the locations of ice shelf features. I recommend explicitly naming the two basal channels discussed to remove ambiguity (for example, abbreviate the large, more central basal channel DMC (Dotson Main Channel) and the smaller channel at the western shear margin DWC (Dotson Western Channel) or similar). I recommend including a brief section on the morphology of the entire ice shelf, either in the Introduction or beginning of the Results (since a major result of this study is high-resolution elevation maps) in which the features can be identified. They could also be marked on this paper's Fig. 1 with superimposed lines of different colors/weights/styles as in Fig. 1 of Dow et al. (2018, Science Advances – Nansen ice shelf channels). This will make it easier for the reader to orient themselves throughout the paper.

We will name the channels Dotson Main Channel and Dotson Coastal Channel and be consistent with that throughout the paper. We do not agree that high resolution elevation maps are a major result of this study, but rather what we use them for, namely the high resolution basal melt rates. Further, surface depressions in an ice shelf does not necessarily mean high basal melt rates which we also see on Dotson. In the Figure below we have marked an anchor shaped surface feature on the ice shelf which is not associated with high melt rates. These kinds of features are very interesting, especially because we do not know what has formed this feature, but it is out of the scope of this paper. Therefore, we also believe that adding a section on the morphology of the ice shelf will reduce the readability of the paper, since our focus is not on surface features, but on melt features from which some happen to be explained by surface features.
Since we have decided to exclude the part about the Dotson Coastal Channel from the manuscript we only discuss one channel which we call Dotson Main Channel.

- L. 293 Related to comment on L. 186 above – please describe how the Dotson/Crosson border was defined and/or why was the ASAID grounding line is used to define the Dotson ice shelf

We use the Dotson Ice Shelf as a test site to show what BURGEE is capable of, for which the ASAID grounding line is used to define the Dotson/Crosson border. We acknowledge that newer and better grounding line products exist for the Dotson/Crosson ice shelf/shelves, but they do not change the findings of this paper.
No adjustments made in the manuscript based on this comment.

- L. 324 Please specify what the cross section is and what quantity (basal melt, thinning?) has an error of +- 2 m/yr

"resulting error" will be replaced with "basal melt rate difference".
This has been specified throughout the section.

- L. 247 I recommend removing the gradient calculation method types or describing them briefly for clarity

Please refer to the new structure of the methods section, which also includes a move of this section.
We have removed the gradient calculation method types as recommended.

**Reviewer 2 comments:**

L129: Any seasonal variability in the bias?

We did not look into this since the REMA strips availability more or less is constrained to the austral summer period.
No adjustments made in the manuscript based on this comment.

L136: You would probably need a reference to support the statement of lack of velocity change between 2010 and 2017. Figure 3b of Wild et al., 2023 suggest that areas of slowdown and acceleration exists through Dotson during this period, interestingly matching some of the melt patterns observed including the new marginal channel.

It is not entirely clear from the Wild et al., 2023 manuscript how the velocities are obtained. The ITS_LIVE velocities over Dotson are incomplete up until 2013. However, when we calculate the trend of all yearly ITS_LIVE velocities from 2013-2018 and applying roughly the same colour scheme as Wild et al Fig. 3 we see the following pattern:

[Figure]

Here we see an almost uniform pattern of a very slight acceleration. Further, it seems that the velocity changes mentioned in Wild et al happened before our study period began. In the updated manuscript we will add a reference to the Lilien et al., 2018 paper who argues that Dotson was stable: "Dotson, which has maintained its speed despite increasingly high melt rates near its grounding line"
In the updated manuscript we refer to Lilien et al. 2018. We have not made any further adjustments based on this comment.

L254: Same comment as in line 136. How would the velocity change described in Wild et al. translate into divergence? I am curious also whether the coregistration refinement is robust enough to be used to refine the divergence between DEM dates?

As seen in the figure above we do not see the same trend/velocity change as Wild et al. when using all yearly ITS_LIVE velocities from 2013-2018. The trend pattern which we can see in the above pattern has a speckled pattern due to noise in the yearly velocities, noise which is not present to the same extent in the averaged ITS_LIVE product. There are no sharp edges between areas of deceleration and acceleration, which is why the resulting signal in the divergence field due to changing velocities will be limited.

Whether or not the coregistration refinement is robust enough to refine the velocities and thereby also the divergences as well is a good question. However, in our study, the final correction / feature tracking is never larger than 300 m, and often also much less than that. So given the 120 m surface velocity resolution, the extra correction which the feature tracking may provide will most likely be too small to properly affect the divergence. No adjustments made in the manuscript based on this comment.

---

## Referee Report (RR1)

Second review for "REMA reveals spatial variability within the Dotson Melt Channel"

The authors have done an excellent job responding to reviewer comments. I have only minor suggestions for clarification and a few typographical catches, detailed below.

Also, for future reference (I do not think it's necessary to update for this study), there is a newer DTU MDT model available - https://ftp.space.dtu.dk/pub/DTU22/MDT/

**Points of clarification**
L. 143 – Please add a brief justification for the 30m filtering threshold. It seems that this might filter out the advection of some large crevasses, but as the authors indicate later, these are not the focus of the study.
L. 219 – is the "fourth criterion" referenced here described in the previous sentence that begins with "Likewise"? If so, please add "(iv)" somewhere in that sentence.
L. 223 – Please add a brief description of how the "yearly median DEM" is produced. Is this the same as the yearly mosaics shown in Fig. 3 and Fig. 4E?
L. 224 – Please add a brief clarification on how the tri-yearly linear trend is defined. Based on the later references to it and Fig. 6, it seems like the 2010/11-2014/15 map in Fig. 6d, for example, is produced from all possible combinations of yearly mosaics in that time period.
L. 261-264 – Please consider adding a brief statement on accounting for the flow-shifting of ice across the grounding line. Do you consider the vertical component of velocity there as the ice flows down steep slopes?
L. 353 – Please indicate what the distance 10-15 km refers to (I assume it is distance along cross-section B-BB)

**Typographical suggestions**
L. 154 – consider "...related to surface processes, monthly surface mass balance values ($M\_s$ in eq. (4)) are obtained...
L. 164 – "This also means that we have applied..." → "This also means that we apply" for tense consistency
L. 187 – Consider "...regardless of tides etc." → "...regardless of variations in sea level"
L. 193 – I don't think the sentence beginning with "Possible implications..." is needed.
L. 216 – "1 and 2" → "(i) and (ii)" for consistency. Why is criterion (ii) self-referenced here?
L. 217 – "...sufficient amount of REMA..." → "...sufficient number of REMA..."
L. 269 & 282 – add "eq." with "(4)"
L. 278 - → "...DEM mosaics is then obtained..."
L. 309 - → "...eastern zone"
L. 342 - → "Dotson Main Channel" for consistency

---

## Author Response (AR2)

Dear Christian,

Thank you very much!
Below you will find our responses to all technical corrections from reviewer 2.

Best Regards
Ann-Sofie

**Points of clarification**

L. 143 – Please add a brief justification for the 30m filtering threshold. It seems that this might filter out the advection of some large crevasses, but as the authors indicate later, these are not the focus of the study.

We have added the following sentence: "This might filter out the advection of some large crevasses. However, since the REMA mosaic of Dotson is composed of REMA strips from mainly 2015 and 2016 this should mostly affect strips from the early years, if at all."

L. 219 – is the "fourth criterion" referenced here described in the previous sentence that begins with "Likewise"? If so, please add "(iv)" somewhere in that sentence.

Typo left from the original version of the manuscript. "fourth criterion" has been changed to "third criterion".

L. 223 – Please add a brief description of how the "yearly median DEM" is produced. Is this the same as the yearly mosaics shown in Fig. 3 and Fig. 4E?

Have added the following sentence: "The yearly median DEM is the median of all overlapping REMA DEM strips per year, with July 1st defined as the first day of the year."

L. 224 – Please add a brief clarification on how the tri-yearly linear trend is defined. Based on the later references to it and Fig. 6, it seems like the 2010/11-2014/15 map in Fig. 6d, for example, is produced from all possible combinations of yearly mosaics in that time period.

In Section 4.3 on the Eulerian elevation change, the following sentence has been added: This implies that both the tri-yearly and entire study period trends are produced from all strips available in the period considered.

At the end of Section 4.5 on Lagrangian elevation change and basal melt rate the following sentence has been added: "As for the Eulerian elevation change (Sect. 4.5), this implies that both the tri-yearly and entire study period trends are produced from all strips available in the period considered and that some pixels may not have data from all years considered."

L. 261-264 – Please consider adding a brief statement on accounting for the flow-shifting of ice across the grounding line. Do you consider the vertical component of velocity there as the ice flows down steep slopes?

As mentioned in that paragraph and the following paragraph, we use the second displacement (i.e. feature tracking) to align features not well aligned after the surface velocity displacement.

We have chosen not to add a statement on accounting for the flow-shifting of ice across the grounding line, since we believe that it might lead to confusion here due to its complexity and its limited relevance for this particular study. The flow-shifting of ice across the grounding line is not straight forward because the grounding line, in reality, is a grounding zone which is constantly influenced by the dynamic ocean. Therefore, we believe that adding

a brief statement to something which instead needs an entire study, may lead to confusion and disturb the flow of the paper.

L. 353 – Please indicate what the distance 10-15 km refers to (I assume it is distance along cross-section B-BB)

(10-15 km) has been changed to (distances 10-15 km of the B-BB cross section)

**Typographical suggestions**

L. 154 – consider "...related to surface processes, monthly surface mass balance values ($M\_s$ in eq. (4)) are obtained...

Changed

L. 164 – "This also means that we have applied..." → "This also means that we apply" for tense consistency

Changed

L. 187 – Consider "...regardless of tides etc." → "...regardless of variations in sea level"

Changed

L. 193 – I don't think the sentence beginning with "Possible implications..." is needed.

Sentence deleted

L. 216 – "1 and 2" → "(i) and (ii)" for consistency. Why is criterion (ii) self-referenced here?

Changed to "(i)". The self-referencing of criterion (ii) is a typo and has been removed.

L. 217 – "...sufficient amount of REMA..." → "...sufficient number of REMA..."

Changed

L. 269 & 282 – add "eq." with "(4)"

Changed

L. 278 - → "...DEM mosaics is then obtained..."

Changed

L. 309 - → "...eastern zone"

Changed

L. 342 - → "Dotson Main Channel" for consistency

Changed